# Dynamics of nitrogen and phosphorus accumulation and their stoichiometry along a chronosequence of forest primary succession in the Hailuogou Glacier retreat area, eastern Tibetan Plateau

Danli Yang[1], Ji Luo[2], Peihao Peng[1]*, Wei Li[2], Wenbo Shi[3], Longyu Jia[1], Yongmei He[2,4]

1 College of Earth Sciences, Chengdu University of Technology, Chengdu, Sichuan, China, 2 Key Laboratory of Mountain Surface Processes and Ecological Regulation, Institute of Mountain Hazards and Environment, Chinese Academy of Sciences, Chengdu, Sichuan, China, 3 Tourism and Urban-Rural Planning College, Chengdu University of Technology, Chengdu, Sichuan, China, 4 University of Chinese Academy of Sciences, Beijing, China

* pengpeihao@cdut.edu.cn

**Data Availability Statement:** All relevant data are within the manuscript and its Supporting Information files.

## Abstract

As the two limiting nutrients for plants in most terrestrial ecosystems, nitrogen (N) and phosphorus (P) are essential for the development of succession forests. Vegetation N:P stoichiometry is a useful tool for detecting nutrient limitation. In the present work, chronosequence analysis was employed to research N and P accumulation dynamics and their stoichiometry during forest primary succession in a glacier retreat area on the Tibetan Plateau. Our results showed that: (1) total ecosystem N and P pools increased from 97 kg hm$^{-2}$ to 7186 kg hm$^{-2}$ and 25 kg hm$^{-2}$ to 487 kg hm$^{-2}$, respectively, with increasing glacier retreat year; (2) the proportion of the organic soil N pool to total ecosystem N sharply increased with increasing glacier retreat year, but the proportion of the organic soil and the vegetation P pools to the total ecosystem P was equivalent after 125 y of recession; (3) the N:P ratio for tree leaves ranged from 10.1 to 14.3, whereas the N:P ratio for total vegetation decreased form 13.3 to 8.4 and remained constant after 35 y of recession, and the N:P ratio for organic soil increased from 0.2 to 23.1 with increasing glacier retreat. These results suggested that organic soil N increased with increasing years of glacier retreat, which may be the main sink for atmospheric N, whereas increased P accumulation in vegetation after 125 y of recession suggested that much of the soil P was transformed into the biomass P pool. As the N:P ratio for vegetation maintained a low level for 35–125 y of recession, we suggested that N might be the main limiting element for plant growth in the development of this ecosystem.

**Funding:** This research was financially supported by the Second Tibetan Plateau Scientific Expedition and Research Program (STEP) (Grant No. 2019QZKK0307), the National Natural Science Foundation of China (Grant No. 41771062) and Key Technical Talents Program of Chinese Academy of Sciences(Grant No. 2016065).

**Competing interests:** The authors have declared that no competing interests exist.

## Introduction

Nitrogen (N) and phosphorus (P) deposition is considered a major contributor to global climate change [1]. Fossil fuel burning, vast deforestation, and fertilizer consumption have increased atmospheric deposition since the industrial and agricultural revolutions [2–4]. Currently, about 60% of atmospheric N is sourced from anthropogenic activities [5], which makes forests one of the main N sinks for global anthropogenic N input [6–8]. The accumulation of N by forests has received much attention in previous studies, and the chronosequence approach provides a key opportunity for estimating the dynamics of N pools and the N accumulation rate in different stages of succession [9–11]. Increases in the vegetation N pool have been reported by many studies on stand development forests [12–15]. However, soil N pools either increased [16, 17] or decreased [18, 19] with forest succession development. These contrary results for the N accumulation in soil were mainly found during secondary succession, indicating that the influence of previous vegetation and soil may not be easily excluded. However, primary succession develops forest communities in previously non-vegetated areas, which are typically not affected by human or historical factors [20–23]. Therefore, studying primary succession will further our understanding of the dynamics of N accumulation under natural conditions.

Mineral weathering in parent rock material is the principal source of P in terrestrial ecosystems [24–26], which means that, against the background of increasing global atmospheric N concentration, available P, which is essential for plant growth, may become a controlling factor for forest primary succession. Therefore, the chemical weathering of minerals [27–31] and P bioavailability [25, 32, 33] have been studied extensively during primary succession. However, there remains a paucity of studies on P dynamics in vegetation and the total ecosystem during primary succession, especially in glacier retreat areas. As the two limiting nutrients for plants in most terrestrial ecosystems, N and P are fundamental to both individual organisms and entire ecosystems [34, 35], and their stoichiometry is often used to study forest succession and the nutrient supply and demand of ecosystems [36, 37]. Many scholars used the N:P ratio as an indicator of the nutrients that limit plant productivity [38–40]. However, few studies have estimated the N–P relationship with ecosystem components during primary succession, and it remains unclear whether this interaction is reflected in changes in the N:P ratio.

Global warming has been in effect since the Little Ice Age, and has accelerated the melting of mountain glaciers. Consequently, the global glacier retreat area has been expanding [41, 42]. The new ice-free areas that have resulted from glacial retreat provide an opportunity to study forest primary succession due to the colonization of these areas by terrestrial plants. Since the end of the 19[th] century, studies on forest primary succession in glacial retreat areas have been carried out in the Glacier Bay (Alaska) [20, 43–45], Damma Glacier (Switzerland) [33, 46, 47], and the Franz Joseph Glacier (New Zealand) [48, 49]. The Hailuogou Glacier retreat area in the Gongga Mountain region has also developed a soil and forest primary succession chronosequence, and has not been greatly disturbed by human activities. Thus, it is an ideal site to study the dynamics of elements during ecosystem development at the centennial scale. According to Luo et al. [50, 51], who measured soil respiration at different succession stages along this sequence, the highest values were found from April to September. He et al. [52] reported that the vegetation C pools increased from 5 t hm$^{-2}$ to 346 t hm$^{-2}$ along the same sequence. Yang et al. [11] found that ecosystem N pools increased with succession but because only broad-leaved forest stages within 60 y of glacial retreat were studied, the change in N along the complete succession sequence was not reflected in the results, especially for the coniferous forest formed after 125 y glacial retreat. Jiang et al. [53], who estimated the N:P ratios in leaves, highlighted that the limiting factor for plant growth shifted from N to P over one century of succession. However, this conclusion was doubted by Zhou et al. [54], who

considered that the P limitation for plant growth at the last stage (coniferous forest stage) of this succession sequence was unlikely due to the soil P supply [32, 55]. Therefore, further in-depth research on the N and P pools of this succession sequence is needed for understanding their relationship in the ecosystem and the impact of their limitation along this forest primary succession. In the present study, the accumulation dynamics and stoichiometry of N and P were estimated in a complete forest primary succession sequence of a glacier retreat area on the eastern Tibetan Plateau. The objectives of our research were to: (1) estimate the N and P pools of each ecosystem component; (2) estimate the proportions of N and P in the vegetation and organic soil to the total ecosystem along the succession sequence; and (3) assess the nutrient limiting factors affecting the development of this ecosystem by describing the N:P ratio.

## Materials and methods

### Study site

The Hailuogou Glacier located on the eastern Tibetan Plateau is the largest valley glacier on the east slope of Gongga Mountain (101˚30'–102˚15' E, 29˚20'–30˚20' N) (Fig 1). The annual average temperature and precipitation is ~4˚C and ~2000 mm, respectively [50]. The relatively mild and humid climate near the glacier has prevented glacial advancement in the past 100 y and promoted rapid moraine colonization by plants. A complete forest primary succession sequence of approximately 2 km was formed in the Hailuogou glacier retreat area at an altitude of 2800–2970 m. In the present study, each glacier retreat year was determined by the ecesis interval of pioneering tree species (i.e., time between glacier retreat and tree seedling germination) and the maximum tree age in the glacier retreat area [56]. The glacier retreat period was from 1890 to 2015. Site S1 represents 15 y of glacier retreat and was invaded by pioneer herbs and trees, including *Hippophae rhamnoides* Linn, *Salix* spp., *Populus purdomii* Rehd, and several other leguminous herbs. Sites S2 to S4 represent glacier retreat after 35 y, 45 y, and 57 y, respectively. During this period, *P. purdomii* became the predominant species owing to its fast growth rate and high photosynthetic rate, which meant it was able to outcompete *H. rhamnoides* and *Salix* spp. Sites S5 and S6 represent glacier retreat after 85 y and 125 y, respectively. *P. purdomii* was gradually replaced by *Abies fabri* (Mast.) Craib during this period [57].

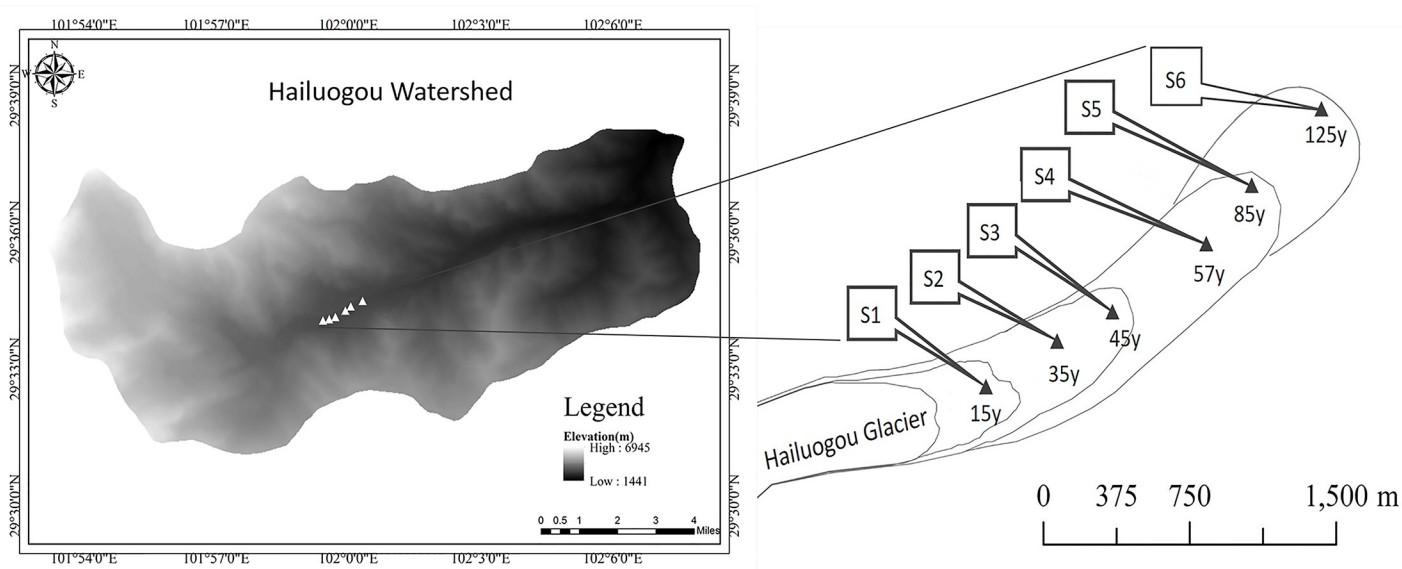

**Fig 1. Sampling sites at Hailuogou glacier retreat area.** S1–S6 are the sampling sites.

## Vegetation sample collection

Our research station is in the Gongga Mountain Nature Reserve, and the current study was carried out on the fixed sample plot set up by the research station. We work in close coordination with the management department of the Reserve. Our work is also part of the monitoring and management of the Reserve. The general observation experiment did not need to be approved. Therefore, all field sampling mentioned in this study was permitted.

We set up six sites in this glacier retreat area in May 2015 (Table 1). The sampling protocol for the vegetation biomass has been described in detail elsewhere [56, 57]. Briefly, six sampling sites were chosen for vegetation sampling based on the glacier retreat year (15 y, 35 y, 45 y, 57 y, 85 y, and 125 y). At each site, three quadrats of 10 m × 10 m were established. All trees with a diameter at breast height (DBH; 1.3 m height aboveground) of >2 cm were inventoried in each quadrat. The species name, DBH, height, and geographical coordinates were recorded in each quadrat. In each quadrat of 10 m ×10 m, we also established a quadrat of 5 m × 5 m to harvest the shrub biomass, and a quadrat of 1 m × 1 m to harvest herb and moss biomass. The shrub, herb, and moss biomass was measured using destructive sampling, and the dried weight of each understory vegetation layer was taken as its biomass. Further data on biomass were retrieved from Yang's [57] report, and the present research is an extension of that work.

We collected tree materials (leaves, branches, stems, bark, and roots) and understory vegetation materials (mixed samples of each understory vegetation layer), and all materials were oven dried at 60°C to a constant weight. The dried samples were ground to a fine powder and used to measure the nutrient concentration. The N concentration was measured by an element analyzer (Vario Macro Cube C, Elementar, Germany). The P concentration was measured by inductively coupled plasma optional emission spectroscopy (ICP-OES 7000DV, Perkin Elmer, USA).

## Litter and soil samples collection

According to terrain, slope, and vegetation distribution, we established a 0.5 m × 0.5 m soil pit in the plot in each quadrat of 10 m × 10 m. The litter (undecomposed) was mainly composed of leaves, and its biomass and nutrient concentration were determined in the same way as for the understory vegetation material. After harvesting the litter samples, a soil profile was hand dug in each plot. From top to bottom soil, an O layer (organic soil), C-layer soil (soil parent

**Table 1. Investigation of vegetation and soil at different glacier retreat years.**

| Sample site | | S1 | S2 | S3 | S4 | S5 | S6 |
|---|---|---|---|---|---|---|---|
| Glacier retreat year | | 15 y | 35 y | 45 y | 57 y | 85 y | 125 y |
| Dominant trees | | H. rhamnoides, Salix spp., P. purdomii | H. rhamnoides, Salix spp., P. purdomii | P. purdomii (half-mature), A. fabri | P. purdomii (mature), A. fabri | P. purdomii A. fabri, | A. fabri, |
| Total living biomass (t hm$^{-2}$) | | 7 (1) | 121 (13) | 199 (22) | 225 (10) | 291 (24) | 366 (19) |
| pH of organic soil | | 6.9 (0.3) | 6.4 (0.5) | 5.5 (0.5) | 5.6 (0.4) | 5.2 (0.4) | 4.4 (0.4) |
| Thickness (cm) | Oe layer | 0.8 (0.1) | 1.1 (0.2) | 1.4 (0.2) | 1.8 (0.3) | 2.3 (0.9) | 3.6 (0.8) |
| | Oa layer | -- | 1.8 (0.7) | 2.6 (0.2) | 3.8 (0.9) | 4.6 (1.2) | 5.4 (1.2) |
| Bulk density (g cm$^{-3}$) | Oe layer | 0.13 (0.01) | 0.12 (0.02) | 0.11 (0.06) | 0.12 (0.07) | 0.19 (0.03) | 0.35 (0.05) |
| | Oa layer | -- | 0.31 (0.11) | 0.38 (0.10) | 0.33 (0.09) | 0.25 (0.06) | 0.30 (0.09) |

Data shown as means with standard deviation in parentheses.

layer), and bedrock were found. The absence of a distinct A (mineral) layer has been attributed to the short period of vegetation succession [56]. The O layer was further divided into Oe (intermediate decomposed organic layer) and Oa (highly decomposed organic layer) [58]. We collected soil samples and measured their thickness, and a separate set of samples for soil bulk density were collected with a cutting ring in each soil layer. The soil samples were air dried at room temperature (~15°C) by placing the samples on kraft paper and then measuring the moisture and calculating the bulk density. Each soil sample was ground in an agate mortar and sieved through a 200-mesh sieve. Owing to the N concentration in C layer being extremely low and having no significant difference in this succession sequence, we only estimated the N and P pools in the organic soil. The N concentration of soil was measured using the semimicro Kjeldahl method. The P concentration of soil was measured by inductively coupled plasma optional emission spectroscopy (ICP-OES 7000DV, Perkin Elmer, USA).

## Statistical analysis

The N and P concentration in tree leaves was based on the leaf biomass of each dominant tree specie at different sites. We calculated the N and P concentration in the leaves of the tree layer as follows:

$$L_{\text{tree}} = \frac{L_H \times B_H + L_S \times B_S + L_P \times B_P + L_A \times B_A}{B_H + B_S + B_P + B_A} \tag{1}$$

where $L$ is N or P concentration in leaves; $B$ is leaf biomass of each dominant tree; $H$, $S$, $P$, and $A$ are *H. rhamnoides*, *Salix* spp., *P. purdomii*, and *A. fabri*, respectively.

The N and P pools of each vegetation layer were estimated by multiplying the N and P concentrations with each biomass component. The soil N and P pool of the Oe and Oa layers can be estimated according to the measured concentration, thickness and the bulk density. In addition, we used the rate of relative N or P change to indicate how the accumulation of N changed with P accumulation in various ecosystem components:

$$R = \frac{(S_i/S_{i-1})-1}{T_i - T_{i-1}} \tag{2}$$

where $R$ is the rate of relative N or P change; $S$ represents the N or P pool of each ecosystem component at each site; $T$ represents the year of glacier retreat in each site; and $i$ is sampling site ($i = 1$–$6$). How N accumulation changed with P accumulation along the succession sequence was determined by comparing the regression slopes for relative N and P change rates with 1 in the 95% confidence intervals [11, 12]. The relationship between N and P for each ecosystem component contributed to the overall understanding of N–P interactions along the succession gradient.

All statistical analyses were conducted using SPSS 21.0 software. One-way ANOVA was used to examine the statistical significance in N pool, P pool, and N:P ratio at a threshold $P$ value of 0.05 in different sites along this succession sequence. The relationships between the relative change rates of N and P were analyzed by linear regression. The logarithmic model was used to simulate the relationship between glacier retreat year and N:P ratio. All graphs were made in Origin 2020.

## Results

### N and P pools in each ecosystem component

During this succession sequence, the total ecosystem N pool increased from 97 kg hm$^{-2}$ to 7186 kg hm$^{-2}$. The total vegetation N pool also showed an increased trend from 93 kg hm$^{-2}$ to

**Table 2. N and P pools in various ecosystem components.**

| Sampling site | | S1 | S2 | S3 | S4 | S5 | S6 |
|---|---|---|---|---|---|---|---|
| Glacier retreat year | | 15 y | 35 y | 45 y | 57 y | 85 y | 125 y |
| **Tree** | N pool (kg hm$^{-2}$) | 75 (10) [e] | 702 (99) [d] | 791 (103) [cd] | 938 (70) [c] | 1381 (95) [b] | 1864 (123) [a] |
| | P pool (kg hm$^{-2}$) | 6 (1) [e] | 74 (6) [d] | 84 (8) [d] | 107 (4) [c] | 183 (13) [b] | 199 (11) [a] |
| **Shrub** | N pool (kg hm$^{-2}$) | 0.3 (0.1) [d] | 0.6 (0.1) [d] | 32 (2) [cd] | 75 (18) [c] | 148 (21) [b] | 319 (58) [a] |
| | P pool (kg hm$^{-2}$) | 0.01 (0.04) [c] | 0.1 (0.03) [c] | 3 (0.2) [c] | 8 (2) [b] | 11 (2) [b] | 22 (4) [a] |
| **Herb** | N pool (kg hm$^{-2}$) | 4 (0.1) [b] | 8 (2) [b] | 7 (1) [b] | 24 (5) [a] | 8 (2) [b] | 23 (7) [a] |
| | P pool (kg hm$^{-2}$) | 0.2 (0.1) [d] | 0.9 (0.2) [c] | 1 (0.1) [c] | 4 (1) [a] | 0.6 (0.1) [c] | 2 (0.2) [b] |
| **Moss** | N pool (kg hm$^{-2}$) | 1 (0.3) [d] | 7 (2) [cd] | 23 (2) [c] | 19 (2) [cd] | 151 (23) [a] | 85 (6) [b] |
| | P pool (kg hm$^{-2}$) | 0.1 (0.02) [d] | 1 (0.3) [cd] | 2 (0.3) [c] | 2 (0.4) [c] | 11 (2) [a] | 7 (1) [b] |
| **Litter** | N pool (kg hm$^{-2}$) | 13 (1) [d] | 32 (2) [c] | 41 (1) [a] | 43 (1) [a] | 42 (1) [a] | 35 (1) [b] |
| | P pool (kg hm$^{-2}$) | 0.5 (0.01) [c] | 1 (0.1) [b] | 2 (0.1) [b] | 2 (0.1) [a] | 1 (0.1) [b] | 1 (0.1) [b] |
| **Oe layer** | N pool (kg hm$^{-2}$) | 4 (0.1) [d] | 273 (23) [c] | 255 (11) [c] | 434 (62) [b] | 503 (69) [b] | 1919 (231) [a] |
| | P pool (kg hm$^{-2}$) | 18 (4) [c] | 19 (4) [c] | 20 (6) [c] | 27 (5) [bc] | 31 (6) [b] | 126 (3) [a] |
| **Oa layer** | N pool (kg hm$^{-2}$) | -- | 1085 (59) [d] | 1581 (356) [c] | 2065 (297) [b] | 2185 (111) [b] | 2941 (35) [a] |
| | P pool (kg hm$^{-2}$) | -- | 84 (14) [b] | 103 (28) [ab] | 130 (12) [a] | 87 (7) [b] | 130 (7) [a] |

Data shown as means with standard deviation in parentheses. Different letters in the same row indicate significant difference among N or P pools among different sites ($P < 0.05$).

2326 kg hm$^{-2}$ (Table 2). The N pool in trees and shrubs exhibited the same increasing pattern with increasing glacier retreat year (trees, $R^2 = 0.9$; shrubs, $R^2 = 0.9$, $P < 0.01$). However, the N pool in herbs and litter were the highest after 57 y of recession; in mosses, the N pool was the highest after 85 y of recession. In the vegetation N pool, 80% of the N was stored in the trees. Additionally, the N pool in organic soil also increased from 4 kg hm$^{-2}$ to 4860 kg hm$^{-2}$ with increasing glacier retreat year ($R^2 = 0.9$, $P < 0.01$). Furthermore, a change in N distribution was detected in vegetation and organic soil (Fig 2A). After 15 y of recession, 97% of the total ecosystem N was stored in vegetation, and only 3% of this amount was found in organic soil.

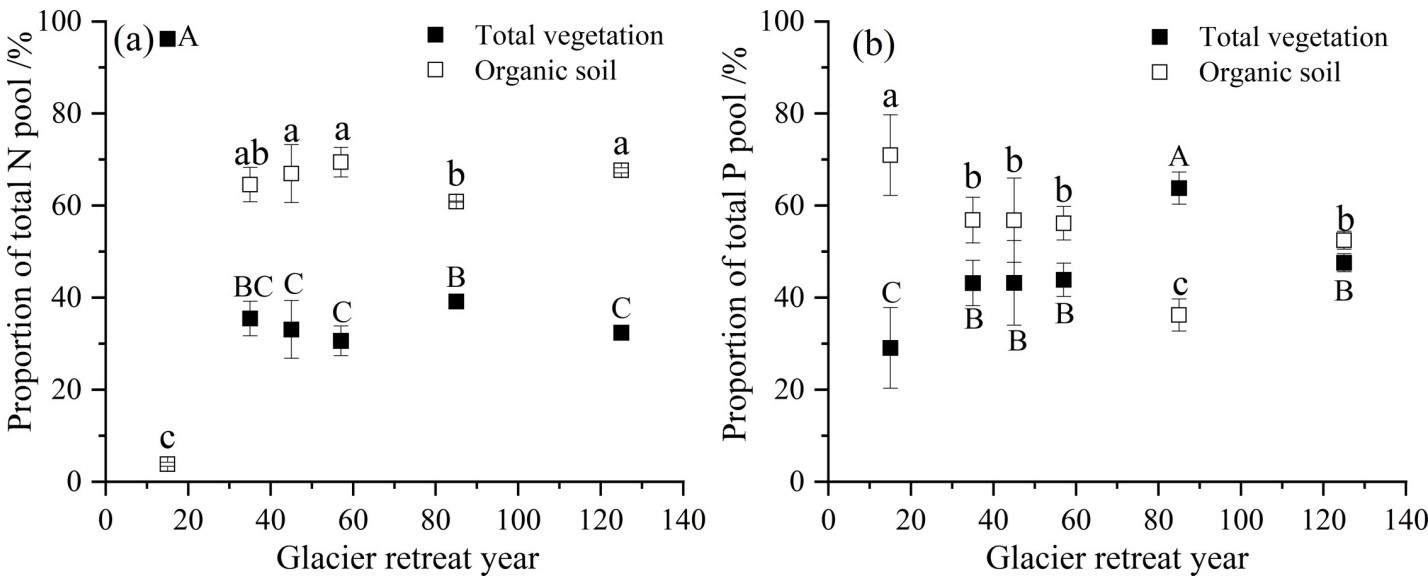

**Fig 2. Proportion of N and P pool in vegetation and organic soil.** Capital letters indicate significant differences in the proportion of N or P in total vegetation at different sites; lowercase letters indicate significant differences in the proportion of N or P in organic soil at different sites ($P < 0.05$).

However, after 125 y of recession, organic soil N comprised 68% of the total ecosystem N, and only 32% was stored in vegetation. Thus, the proportion of organic soil N pool to total ecosystem N increased with the increase in glacier retreat year.

The total ecosystem P pool increased from 25 kg hm$^{-2}$ to 487 kg hm$^{-2}$ at this succession sequence. The P pool in vegetation increased from 7 kg hm$^{-2}$ to 231 kg hm$^{-2}$. The P pool in trees and shrubs also increased with increasing glacier retreat year (trees, $R^2 = 0.9$; shrubs, $R^2 = 0.9$, $P < 0.01$). However, the P pool in herbs and litter were the highest after 57 y of recession; and in mosses, the P pool was the highest after 85 y of recession. The tree P accounted for approximately 89% of the total vegetation P pool. The organic soil P pool also exhibited an increased trend along this succession sequence from 18 kg hm$^{-2}$ to 256 kg hm$^{-2}$. After 125 y of recession, the proportion of P pool in vegetation was equivalent to that in organic soil (Fig 2B).

## N–P correlations in each ecosystem component

There was a positive correlation between the rates of relative N and P changes in tree, shrub, total vegetation, and organic soil (Fig 3). The slope between the rates of relative N and P changes was <1 for shrub (0.86 ± 0.04) and organic soil (0.01 ± 0.001). These findings indicated that the rate of relative N change in these ecosystem components was higher than that of relative P change. However, tree (1.33 ± 0.02) and total vegetation (1.46 ± 0.02) yielded slopes >1, indicating the rate of relative P change was higher than the relative N change in the tree layer. Additionally, the rates of relative N and P changes in each ecosystem component was highest in S2~S3 sites at early stage.

## N:P stoichiometry in vegetation and organic soil

The N:P ratio (N concentration: P concentration) of leaves in four dominate tree species were different (Table 3). The N:P ratio of leaves was the highest in *H. rhamnoides*, whereas that of *Salix* spp. was the lowest. The N:P ratio of leaves in *P. purdomii* was the highest after 35 y of recession, and showed a decreasing trend with increasing glacier retreat year. The N:P ratio of leaves in *A. fabri* remained constant at around 10. From the whole tree layer, the N:P ratio of leaves showed a trend of increasing initially and then decreasing. Except for that at the site of 45 y of recession, the N:P ratio of leaves in trees were less than 14.

Furthermore, this work mainly investigated the dynamic changes in N and P pools in the whole primary succession, and we discuss the relationship between N and P in the ecosystem. When studying the N:P stoichiometry of ecosystems, the pools of N and P should be carefully considered [35]. The N:P ratio (N pool: P pool) for total vegetation ranged from 8.4 to 13.3 and remained constant after 35 y of recession ($P > 0.05$; Fig 4). Conversely, the N:P ratio for organic soil increased from 0.2 to 23.1 with increasing glacier retreat year ($R^2 = 0.9$, $P < 0.05$; Fig 4), and was the highest after 85 y of recession. Additionally, the N:P ratio in total vegetation was higher than that in organic soil after 15 y of recession. However, after 35 y of recession, the N:P ratios in organic soil were higher than those in vegetation with increasing glacier retreat year. Therefore, the correlation analysis of the N:P ratio in total vegetation and organic soil showed that there was a significant negative correlation between them ($R^2 = -0.9$, $P < 0.01$).

## Discussion
### Dynamics of N and P in this succession sequence

The accumulation and distribution of nutrients are basic characteristics of forest ecosystems and are also the material basis for maintaining the structure and function of such ecosystems [58]. In the present study, we found that total vegetation N increased along the succession

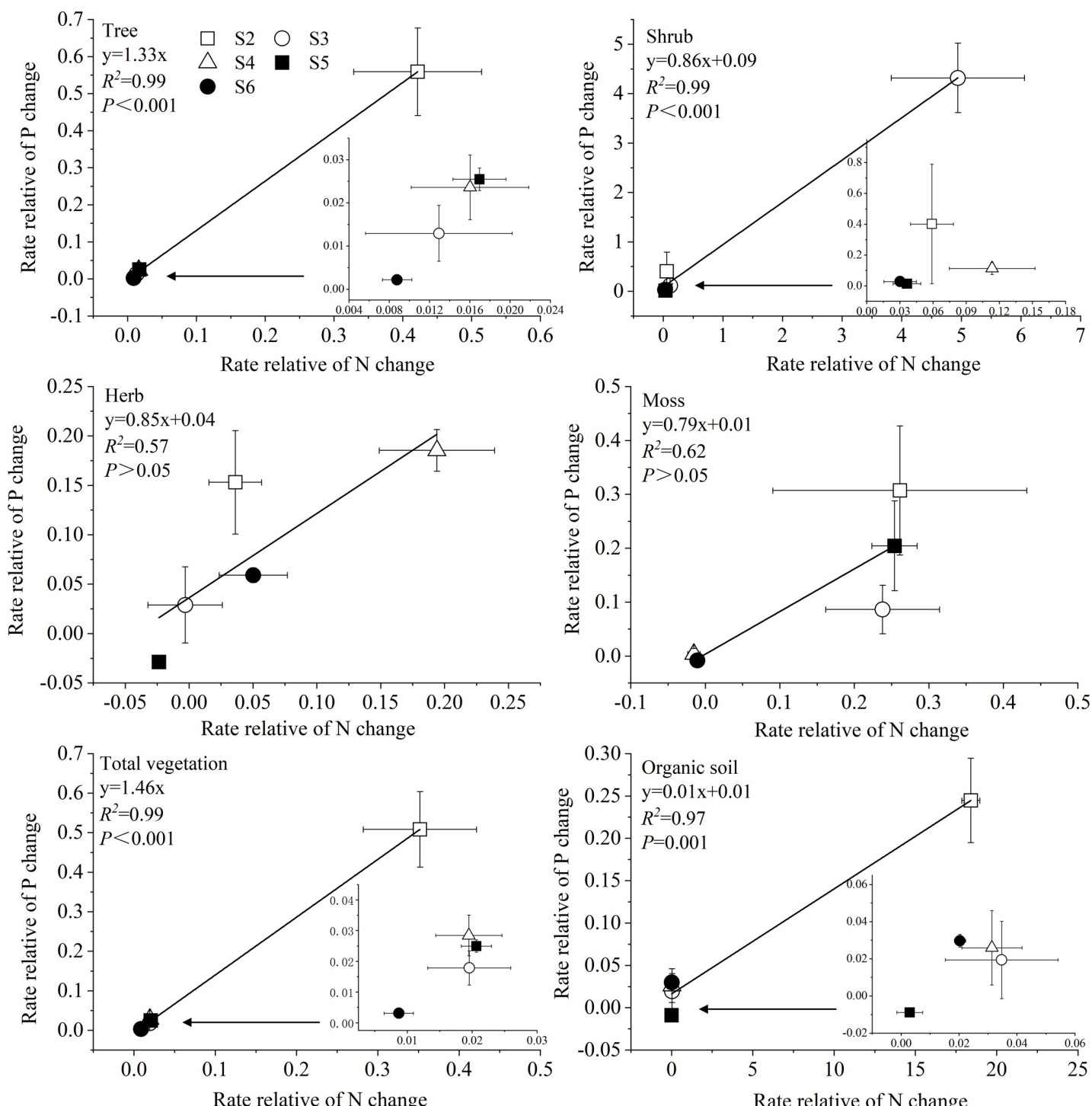

**Fig 3. Relationship between the rate of relative N change and rate of relative P change in each ecosystem component.**

sequence, and 80% of that increase was mainly from trees. Yang et al. [12] reported similar results in >100 forest stand developments. Some studies, such as those on *Eucalyptus* [59, 60] and spruce forest chronosequences [18], also reported similar results. The increased vegetation N pool might have resulted from the gradually increasing biomass in this succession sequence.

**Table 3. N:P ratio of leaves in different species and tree layers.**

| Glacier retreat year | N:P ratio in leaves | | | | |
|---|---|---|---|---|---|
| | *H. rhamnoides* | *Salix* spp. | *P. purdomii* | *A. fabri* | Tree |
| **15 y** | 19.6 | 4.2 | 13.7 | -- | 11.0 |
| **35 y** | 18.2 | 5.0 | 15.4 | -- | 10.1 |
| **45 y** | 17.8 | 10.1 | 14.8 | 10.5 | 14.3 |
| **57 y** | 14.9 | 8.1 | 12.8 | 10.0 | 12.7 |
| **85 y** | -- | -- | 12.6 | 10.4 | 10.4 |
| **125 y** | -- | -- | -- | 10.5 | 10.5 |

Additionally, some studies have shown that the N pool in organic soil non-significantly changed with the age of the sequence [12] or was even lower in the older site [18, 19], which may have resulted from decreasing litter biomass and pedoturbation. In the present study, the

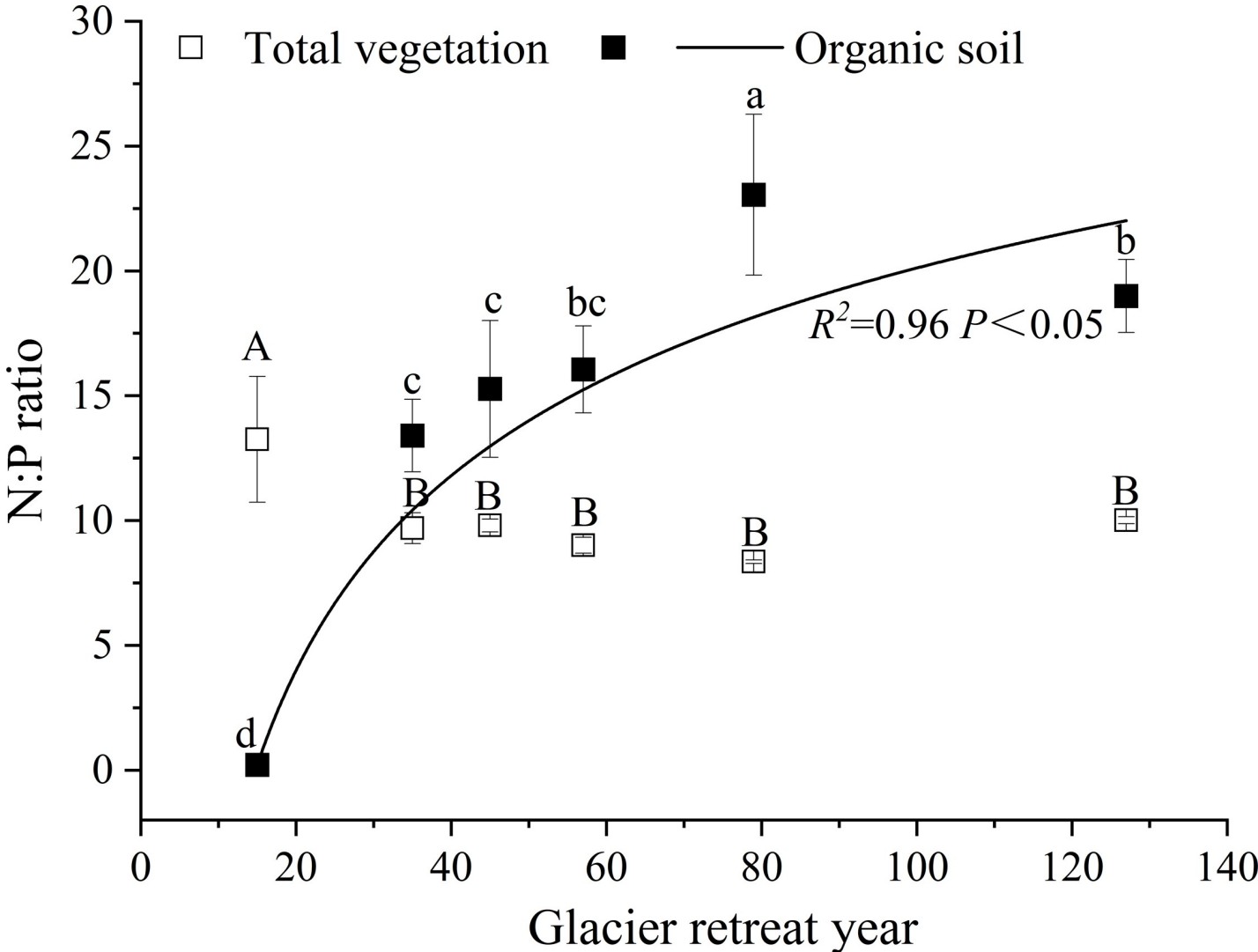

**Fig 4. The N:P ratio in vegetation and organic soil.** Capital letters indicate significant differences in the N:P ratio of vegetation at different sites. Lowercase letters indicate significant differences in the N:P ratio of organic soil at different sites ($P < 0.05$).

N accumulation in the organic soil was positively correlated with glacier retreat year. This was especially evident in the oldest site of 125 y where the organic soil stored more than twice as much N as the vegetation. This larger proportion in the organic soil versus vegetation indicated that the main sink for atmospheric N is soil [61, 62]. Furthermore, the increased organic soil N pool may have also resulted from biological N fixation and atmospheric N deposition [63]. This primary succession sequence formed in the glacier foreland, and the concentration of available N in moraine clay was 0 [57]. However, after 35 y of recession, the organic soil N pool increased to 1358 kg hm$^{-2}$, which shows the importance of leguminous plants [43, 48, 49], such as *Astragalus adsurgens* and *H. rhamnoides*, on N accumulation and soil development in this sequence. With leguminous plants gradually withdrawing from the community, N accumulation was also detected in organic soil at 41 kg hm$^{-2}$ y$^{-1}$ with increasing glacier retreat year. However, the total atmospheric N deposition rate was approximately 8 kg hm$^{-2}$ y$^{-1}$ for this area [64], which showed that the N accumulation rate in the organic soil was higher than atmospheric N deposition rate. Thus, the rate of N deposition was not sufficient to explain the large increase in organic soil N. Many scholars have found that some non-leguminous plants could form nodules and fix N with *Frankia* bacteria of actinomycetes in soil, which is the main sources of N in temperate and boreal forests. For example, the rate of N fixation of *Alnus* Mill. (Actinorhizal plant) is 12–200 kg hm$^{-2}$ y$^{-1}$, which is higher than that of *Hippophae* L. (27–179 kg hm$^{-2}$ y$^{-1}$) [65, 66]. In addition, the cycling rate of soil microorganisms is rapid, and the residues after their death provide rich organic N [67, 68]. Therefore, even if there are no leguminous plants at the older sites in this sequence, the organic soil can also continuously accumulate N.

Total ecosystem P also increased with increasing glacier retreat year. Although the organic soil P pool was larger than that in total vegetation after 15 y of recession, the vegetation P pool increased and was equivalent to that in organic soil after 125 y of recession. Wu [25] reported that the bioavailable P in soil increased rapidly after 30 y of glacier retreat in this sequence. Additionally, the development of fir and spruce in this sequence decreased the pH of soil. This can promote the dissolution of some silicate minerals and promote the production of bioavailable P to meet the needs of different vegetation [32, 69]. Smits [70] reported that material secreted by ectomycorrhizal fungi on the roots of coniferous species can dissolve apatite, suggesting superior acquisition of P in poor soils. All these reports support our finding that much of the organic soil P at the older sites was transformed into the biomass P pool. Especially after 85 y of recession, the P pool of the total vegetation even exceeded that of the organic soil when coniferous species were dominant.

We also found that after 125 y of soil development, the N and P pools in organic soil of this sequence were much lower than those in the surface soil of zonal vegetation dark coniferous forest (N, 7882 kg hm$^{-2}$; P, 2122 kg hm$^{-2}$) [71]. This result indicated that, although the climax community was formed after 125 y of recession, the soil in this chronosequence was still in the "early stage" [54]. Therefore, the dynamics of N and P accumulation in soil need to be further studied.

### N:P stoichiometry and nutrient limitation

Previous studies have highlighted that N:P stoichiometry plays a fundamental role in the structure and functioning of communities and can be used as an indicator of the nutrients that limit plant productivity [38, 39, 72]. Generally, a N:P ratio <14 in leaves indicates that plant growth is limited by N, whereas a N:P ratio >16 indicates that plant growth is limited by P. When the N:P ratio is between 14 and 16, plant growth is simultaneously limited by both N and P [73]. Güsewell [39] also proposed a N:P ratio at the vegetation level (<10 for N

limitation; >20 for P limitation). However, the N:P ratio in different study areas, species, and ecosystems might vary greatly. Therefore, most scholars accept the view of ecosystems being N-limited at a low N:P ratio and P-limited at a high N:P ratio.

In the present study, the N:P ratio of leaves in the tree layer ranged from 10.1 to 14.3. Except for the site with 45 y of recession, the N:P ratio of tree leaves was lower than 14. However, Jiang et al. [53] reported that the community-weighted mean N:P ratios ranged from 8.2 to 20.1 in leaves and highlighted that the limitation factor for plant growth shifted from N to P. Our data do not support this conclusion. First, Jiang et al. [53] only reported the maximum and minimum values of the N:P ratio in leaves, therefore excluding the change in the N:P ratio of leaves in the whole succession sequence. It is not accurate to determine N or P limits only from these two values. Moreover, Jiang et al. [53] reported that the N:P ratio of leaves in the coniferous forest at late stage (20.1) was much higher than that in the broad-leaved forest at early stage (8.2). In most studies, the N:P ratio of leaves in the broad-leaved forest was higher than that of coniferous forest leaves [74], and that of deciduous woody forest was higher than that of evergreen woody forest [39]. Therefore, the calculated N:P ratios presented here are more accurate. Finally, at early stage, especially in the 3–12 y of glacial retreat reported by Jiang et al. [53], the vegetation was mainly dominated by N-fixation plants, such as *H. rham-noides* and *A. adsurgens*, and the N concentration in their leaves was significantly higher than that of other species (S3 File). This might have led to the high N:P ratio of leaves in this period. However, the N:P ratio of leaves calculated by Jiang et al. [53] at this stage was only 8.2. Over-all, our results reflect the changes in the N:P ratio of the leaves of different species and of the entire tree layer along the succession sequence. According to the threshold of N:P nutrient lim-itation, the growth of trees might be limited by N supply.

Biomass N:P stoichiometry reflects the balance between the uptake and loss of N and P [39]. When studying the N and P stoichiometry of ecosystems, the pools of N and P should be carefully considered. Therefore, a given N:P ratio may refer to total nutrient pools (N pool: P pool) [35]. Three factors regarding the ecosystem-level of N:P ratios may help establishing N as limiting nutrient. First, the mass ratio of the leaf only reflects the allocation of N and P at the organ-level, while the pool ratio of N:P reflects the allocation of both nutrients at the ecosys-tem-level. Second, the ecosystem-level of N:P ratios can reflect the relationship between N and P in the different ecosystem components and clarify their interactions along the succession. In the present study, we calculated the rates of relative N and P changes of each ecosystem com-ponent based on N and P pools and found that the N:P ratio in vegetation decreased slightly and was lower than that of organic soil. This was due to the higher P accumulation rate com-pared with the N accumulation rate. The influence of accumulation rates between N and P on N:P ratio is not reflected on the mass ratio. Finally, the total vegetation N:P ratio ranged from 8.4 to 13.3 in the present study. After 15 y of recession, the N:P ratio of total vegetation might have peaked in this succession sequence due to the N-fixing plants at this site. As leguminous plants withdrew from the community, the N:P ratio in vegetation dropped to a constant value along this sequence, reflecting the long-term and stable corresponding values of N and P in vegetation to a state of equilibrium. This is the embodiment of ecosystem stability, thus pro-moting the positive succession of vegetation. Therefore, in the long-term development of the ecosystem, N was considered as the main limiting nutrient.

Additionally, previous studies also confirmed that plant growth in Hailuogou succession sequence may be mainly limited by N. First, Zhou et al. [31, 32, 55] carried out much research on soil P in the Hailuogou glacier chronosequence and reported that, at the 120-y-old-site, the $P_{available}$ pool in the organic layer and mineral soils (0–6 cm) was ~5.3 times that of annual plant P requirement; the $P_{mineralization}$ pool (the supply of available P from mineralization of organic P in the organic layer) was ~2.9 times that of the P requirement; the $P_{weathering}$ pool

was 7.5 kg ha$^{-1}$ y$^{-1}$ and was also higher than the P requirement. These results suggested that the current P pools can offer enough P for the growth of plant. Furthermore, although it is difficult to carry out nutrient-addition experiments in this primary succession sequence, a N-addition experiment on *A. fabri* seedings may be helpful to understand nutrient controls on plant growth. Yang et al. [75] conducted such an experiment in the Gongga Mountain observation station, which is ~1 km from the Hailuogou glacier retreat area. They found that the total biomass, leaf dry weigh, leaf mass ratio, leaf N and P concentration, and leaf N:P ratio of the *A. fabri* seedlings increased by 11.3%, 46.7%, 41.4%, 37.3%, 22.3%, and 6.4%, respectively, after 2 years of adding N (50 kg hm$^{-2}$ y$^{-1}$), indicating that the growth of *A. fabri* seedlings was probably limited by N. Therefore, from the results of N:P stoichiometry of vegetation, soil P supply, and N-addition experiments, we suggested that the plant growth in the Hailuogou glacier succession sequence may be limited by N.

## Conclusions

Global climate change may lead to more widespread glacial retreat, making it even more important to thoroughly understand the nutrient dynamics in the areas undergoing primary succession. Our results showed that the N and P pools in each ecosystem component in a glacier retreat area increased with glacier retreat year. However, the proportion of the organic soil N pool to total ecosystem N gradually increased with increasing glacier retreat year and revealed that the organic soil may be main sink for atmospheric N along this succession sequence. In contrast, the increased P accumulation in vegetation indicated that much of the soil P was converted into the biomass P pool. The N:P ratio in the organic soil increased perhaps due to the higher N accumulation rate compared with P, whereas the N:P ratio in total vegetation maintained a constant low level perhaps due to the tree layer having a higher P accumulation rate compared with N. Furthermore, the tree leaves and total vegetation N:P ratios suggested that N would be the main limiting factor for plant growth along this succession sequence. Understanding the dynamics of nutrients during primary succession may be important to promote the recovery of degraded ecosystems and predict forest migration patterns during periods of future climate change.

## Supporting information

**S1 File. Dataset-raw data of meteorology.**
(XLS)

**S2 File. Dataset-raw data of biomass.**
(XLS)

**S3 File. Dataset-raw data of N and P concentrations.**
(XLS)

**S4 File. Dataset-raw data of rate relative N or P change.**
(XLS)

## Acknowledgments

We are truly grateful to Gongga Mountain Nature Reserve for their assistance in field sampling. We thank the Ecological resources & Landscape institute, Chengdu University of Technology for supporting our research. We thank Xun Wang for his help in sample analysis. We gratefully acknowledge Deane Wang and two anonymous reviewers for their comments that significantly improved the manuscript.

## Author Contributions

**Conceptualization:** Ji Luo, Peihao Peng.

**Data curation:** Danli Yang, Yongmei He.

**Formal analysis:** Danli Yang, Ji Luo, Wei Li.

**Investigation:** Danli Yang, Ji Luo, Wenbo Shi, Yongmei He.

**Methodology:** Danli Yang, Wei Li.

**Software:** Longyu Jia.

**Supervision:** Ji Luo.

**Validation:** Peihao Peng.

**Visualization:** Wenbo Shi, Longyu Jia.

**Writing – original draft:** Danli Yang.

**Writing – review & editing:** Danli Yang.

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
