## [Decision Letter · Decision Letter 0]

10 Nov 2020

PONE-D-20-29430

Dynamics of nitrogen and phosphorus accumulation and their stoichiometry along a chronosequence of succession forests in the Hailuogou Glacier retreat area, eastern Tibetan Plateau

PLOS ONE

Dear Dr. Peng,

Thank you for submitting your manuscript to PLOS ONE. After careful consideration, we feel that it has merit but does not fully meet PLOS ONE’s publication criteria as it currently stands. Therefore, we invite you to submit a revised version of the manuscript that addresses the points raised during the review process.

We look forward to receiving your revised manuscript.

Kind regards,

Dafeng Hui, Ph.D.

Academic Editor

PLOS ONE

Journal Requirements:

2. In your Methods section, please provide additional location information of the sampling sites, including geographic coordinates for the data set if available.

4. We note that Figure1 in your submission contain map images which may be copyrighted. All PLOS content is published under the Creative Commons Attribution License (CC BY 4.0), which means that the manuscript, images, and Supporting Information files will be freely available online, and any third party is permitted to access, download, copy, distribute, and use these materials in any way, even commercially, with proper attribution. For these reasons, we cannot publish previously copyrighted maps or satellite images created using proprietary data, such as Google software (Google Maps, Street View, and Earth). For more information, see our copyright guidelines: http://journals.plos.org/plosone/s/licenses-and-copyright.

(1)    You may seek permission from the original copyright holder of Figure 1 to publish the content specifically under the CC BY 4.0 license. 

Additional Editor Comments (if provided):

I now have three reports from expert reviewers. Reviewers have different opinions and recommendations of the manuscript, from Reject, to Major revision and Accept. Reviewer #1questioned the major hypothesis and a lack of data to supported it. Reviewer #3 is more critical and raised several technical issues from the scientific questions to presentation and language. If the authors think that they can address these issues, I'd like to give them an opportunity to do a substantial revision of the manuscript.

Reviewers' comments:

Reviewer's Responses to Questions

**Comments to the Author**

1. Is the manuscript technically sound, and do the data support the conclusions?

Reviewer #1: Yes

Reviewer #2: Yes

Reviewer #3: Partly

2. Has the statistical analysis been performed appropriately and rigorously? 

Reviewer #1: No

Reviewer #2: Yes

Reviewer #3: No

3. Have the authors made all data underlying the findings in their manuscript fully available?

Reviewer #1: No

Reviewer #2: Yes

Reviewer #3: Yes

4. Is the manuscript presented in an intelligible fashion and written in standard English?

Reviewer #1: Yes

Reviewer #2: Yes

Reviewer #3: No

5. Review Comments to the Author

Reviewer #1: The conclusion that N limits plant growth, however, is not fully supported. The authors should more fully defend this assertion, or simply suggest that an N limitation hypothesis is favored by them.

I recommend publication with some substantial revisions. This is information deserving publication with some clarifications.

Please see more detailed Reviewer comments (attached pdf)

Supplementary material link was provided, but I had technical difficulty and could not access.

Reviewer #2: This paper presents new information on a very important topic regarding the nitrogen and phosphorus cycles in regions not previously studied and information on how the cycling of these elements occur in primary succession caused by glacial retreat. The paper has international significance. The authors carefully collected the required information and did the proper statistical tests. I recommend that the paper be published as is.

Reviewer #3: The manuscript entitled “Dynamics of nitrogen and phosphorus accumulation and their stoichiometry along a chronosequence of succession forests in the Hailuogou Glacier retreat area, eastern Tibetan Plateau” focused on exploring the N and P accumulation dynamics and their stoichiometry during forest primary succession in a glacier retreat area on the Tibetan Plateau. However, they didn’t address any the following key scientific issues that they raised. Firstly, I do not agree with you that surface soil N increased with increasing years of glacier retreat, becoming a main N pool, whereas increased P accumulation in vegetation after 125 y of recession indicated that much of the soil P was transformed into the biomass P pool. Obviously, the increasing N:P ratio for surface soil do not support the conclusion of N or P pool. You should provide the solid evidence. Secondly, some fitting results are incredible. For example, the fitting of N:P ratio for vegetation is straight line? The results of this study may be false positive. As a result, I doubt scientific values and the conclusions in this study. Finally, the MS was poorly written in the language and logic, which is needed to be substantially strengthened before submission. The production of the tables and figures are also very unprofessional. For example, Table 2, I can not understand. The authors do need to pay attention to these issues for any scientific article. Based on these issues, I would not recommend it can be accepted by PlOS ONE.

6. PLOS authors have the option to publish the peer review history of their article (what does this mean?). If published, this will include your full peer review and any attached files.

Reviewer #1: **Yes: **Deane Wang

Reviewer #2: No

Reviewer #3: No

---

## [Author Response · Author response to Decision Letter 0]

17 Dec 2020

17-Dec-2020

Dear Editor,

Thank you very much for your valuable advice, which was helpful for improving our manuscript. We have made substantial revisions to the manuscript according to the reviewers' comments, and the amendments are highlighted in red in the revised manuscript. We also have responded to each comment below, including reviewer 1 and reviewer 3, and changed the file format of the supplementary material so that reviewers can check it. We thank the editor and reviewers for giving us the chance to revise the manuscript. We hope that the revision is acceptable and look forward to hearing from you soon. 

Kind regards,

Danli Yang 

First of all, we thank both editor and reviewers for their positive constructive comments and suggestions.

Replies to editor:

1.Please ensure that your manuscript meets PLOS ONE's style requirements, including those for file naming. The PLOS ONE style templates can be found at https://journals.plos.org/plosone/s/file?id=wjVg/PLOSOne_formatting_sample_main_body.pdf;
https://journals.plos.org/plosone/s/file?id=ba62/PLOSOne_formatting_sample _title_authors_affiliations.pdf

Reply: We have checked and revised my manuscript to meet the PLOS ONE style requirements.

2. In your Methods section, please provide additional location information of the sampling sites, including geographic coordinates for the data set if available.

 Reply: We have already provided location information of the sampling sites, including geographic coordinates, and the amendments are highlighted in red in the revised manuscript. The geographic coordinates for Hailuogou glacier (Gongga Mountain) are 101°30' -102°15' E, 29°20' -30°20' N. We revised the text to address this point in line 133.

3.PLOS requires an ORCID iD for the corresponding author in Editorial Manager on papers submitted after December 6th, 2016. Please ensure that you have an ORCID iD and that it is validated in Editorial Manager. To do this, go to ‘Update my Information’ (in the upper left-hand corner of the main menu), and click on the Fetch/Validate link next to the ORCID field. This will take you to the ORCID site and allow you to create a new iD or authenticate a pre-existing iD in Editorial Manager. Please see the following video for instructions on linking an ORCID iD to your Editorial Manager account: https://www.youtube.com/watch?v=_xcclfuvtxQ

Reply: ORCID iD for the corresponding author Peihao Peng: https://orcid.org/0000-0001-7272-8904

4.We note that Figure1 in your submission contain map images which may be copyrighted. All PLOS content is published under the Creative Commons Attribution License (CC BY 4.0), which means that the manuscript, images, and Supporting Information files will be freely available online, and any third party is permitted to access, download, copy, distribute, and use these materials in any way, even commercially, with proper attribution. For these reasons, we cannot publish previously copyrighted maps or satellite images created using proprietary data, such as Google software (Google Maps, Street View, and Earth).For more information, see our copyright guidelines: http://journals.plos.org/plosone/s/licenses-and-copyright. We require you to either (1) present written permission from the copyright holder to publish these figures specifically under the CC BY 4.0 license, or (2) remove the figures from your submission.

Reply: We removed the previous Figure 1 and supplied a replacement figure. The replacement figure is a new one. The figure has not been published anywhere else, and the copyright belongs to us. Therefore, there is no copyright issue.

Figure 1. Sampling sites at Hailuogou glacier retreat area. S1–S6 are the sampling sites.

Replies to reviewer 1: Deane Wang:

Abstract/Introduction:

1. The abstract and introduction should emphasize the same important points. The abstract makes assertions about the role of the N:P ratio in limiting vegetation growth, while the third objective listed at the end of the introduction just list the desire to report on changes in the N:P ratio. It seems like the overriding goal is to better understand nutrient controls on ecosystem development through the description of N:P ratios, with the specific goal of assessing limiting factors.

Reply: One of the goals of this study was to better understand the role of N or P in the control of plant growth in the process of ecosystem development through the N: P ratio of vegetation. To clarify this point, We have revised the abstract and introduction, and the amendments are highlighted in the revised manuscript, such as in lines 41 and 104.

2.This work is part of a series of papers on the Hailuogou glacier retreat area, which should be acknowledged and cited in the Introduction. This helps the reader understand the context of the research, and may also help understand what work was done in connection with related research and what work was conducted specifically to answer the N:P ratio questions.

Reply: Previous researchers have done a lot of work in Hailuogou Glacier Retreat area, such as quantifying vegetation succession process, C accumulation process, soil P form, and effectiveness evaluation. In the introduction, We have summarized related research results, and the new text is highlighted in the revised manuscript. However, the dynamic changes of N and P in the whole ecosystem have not been studied in this primary succession sequence. Although Yang et al. [1] studied the dynamic changes of C and N in the ecosystem, they only studied the broad-leaved forest stage in this succession sequence, and did not study the coniferous forest stage in the late succession stage, so it is incomplete for the forest succession process. Therefore, based on the previous research results, we studied the dynamic process of the complete vegetation succession sequence for N and P. We revised this for clarity in lines 88-99.

METHODS

1.A key paper to describe vegetation methods might be: Yang D, Luo J, She J, Tang R (2015) Dynamics of vegetation biomass along the chronosequence in Hailuogou glacier retreated area, Mt. Gongga. Ecology and Environmental Sciences 24:1843–1850 (in Chinese with English abstract). However, I could not access this paper. If the chronosequence and the biomass sampling are the same, then perhaps this material could be included in the supplementary material. (I could also not access the supplementary material for this paper - Yang et al.).

Reply: The chronosequence and the biomass sampling in the present study were the same as those reported in Yang’ paper[2]. We set up six sampling sites in the Hailuogou glacier retreat area (glacier retreat time: 2000, 1980, 1970, 1958, 1930, and 1890). Specific biomass information has been added to the supplementary material.

2.Currently, the Methods section omits much detail, and if this is because the work was related to other work, this should be described (and cited). If, for example, the sampling was done specifically for this paper, then a great deal more information needs to be provided.

For example, were the tree subsamples and the allometric equations developed from the same trees? If the vegetation was sampled for this research, how were the replicate samples distributed within a site? Were they randomly selected? Were shrub and ground cover samples nested within the tree samples? Were the soil samples nested in any of the other sample locations? The high spatial variability of soils and vegetation is generally a challenge for ecosystem research and some acknowledgement of this and how this team approached this problem is useful for future investigators.

Reply: The sampling of the vegetation biomass has been described in detail elsewhere [2]. Briefly, six sampling sites were chosen for vegetation sampling basing on the glacier retreat time (2000, 1980, 1970, 1958, 1930, and 1890). At each site, three quadrats of 10 m × 10 m were established. All trees with a diameter at breast height (DBH; 1.3 m height aboveground) of >2 cm were inventoried in each quadrat. The species name, DBH, total height, and geographical coordinates were recorded in each quadrat. Based on the information reported by Yang et al.[2], Liu et al.[3] established the dominant tree species (H. rhamnoides, Salix spp., P. purdomii, and A. fabri). Allometric equations are commonly used to estimate tree and stand biomass from easily measured dendrometric variables such as DBH or height. Briefly, an additive allometric equation for tree biomass components against DBH and H was as follows:

In(y) = a + b*In(DBH^3/H)

Detailed discussion of the four tree species in this study can be found in Liu et al. [3].

The shrub, herb, and moss quadrats were nested within the tree quadrat. In each quadrat of 10 m × 10 m, we established a quadrat of 5 m × 5 m to harvest the shrub biomass, and 1 m × 1 m subplots on the forest floor at each site were prepared, and the mosses and grasses were collected.

The soil samples were nested in each quadrat of 10 m × 10 m. We determined the sampling plot of the soil profile at each sampling site according to terrain, slope, and other conditions, and then a 0.5 × 0.5 m subplot at each site was established to sample litter and soil.

We revised this for clarity in lines 130-143.

3.The context of this research is also a bit puzzling relative to:

Yang et al. (2014) Dynamics of carbon and nitrogen accumulation and C:N stoichiometry in a deciduous broadleaf forest of deglaciated terrain in the eastern Tibetan Plateau. Forest Ecology and Management 312: 10–1.

That work indicates its chronosequence location as "This study was conducted in the foreland of the Hailuogou glacier (101.99°E, 29.57°N, 2990 m a.s.l.)" the whole successional chronosequence, the Populus purdomii forest acts as the broadleaf deciduous stage, which is a band in the valley bottom at elevation from 2844 m to 2950 m." This paper (under review) also list the elevation: "... forest primary succession sequence of approximately 2 km was formed in the Hailuogou glacier retreat area at an altitude of 2800– 2970 m “with similar Lat/Long coordinates. If these chronosequences are the same, the Methods and perhaps the Introduction, should indicate this.

Because the C:N stoichiometry over a chronosequence would require essentially the same methods as a N:P stoichiometry, I would think that these were related studies. This should be clarified.

Reply: Yang et al.[1]and the present study investigated the same forest primary succession sequence in the Hailuogou glacier retreat area, but there were many differences. First, The glacier retreat year was determined by time summary of the ecesis interval of pioneering tree species (i.e., time between the glacier retreat and tree seedlings germinating) and the maximum tree age in the glacier retreated areas [2, 4–6], and through tree rings for correction. Yang et al. [1] was based on the terrain age every 10 years, which may not represent the exact year of glacier retreat. 

Second, which is the most important difference, Yang et al. [1] only studied the broad-leaved forest stages of this forest primary succession sequence. However, this forest primary succession sequence in the Hailuogou glacier retreat area is the complete successional chronosequences from the pioneer vegetation (Astragalus souliei Simps, Hippophae rhamnoides Linn) to broad-leaved forest stages (Populus purdomii Rehd, Hippophae rhamnoides Linn, Salix spp.), and then to the climax community (Abies fabri (Mast.) Craib and Picea asperata Mast.). We studied the whole successional chronosequences from the pioneer vegetation to the climax community (including the broad-leaved forest stages). Therefore, Yang et al. [1]and the present study investigated the same area of forest primary succession but not the same successional chronosequences. Yang et al. [1] only studied the broad-leaved forest stage of this forest primary succession (the glacier retreated for about 60 years), whereas we studied the complete forest primary succession sequence (the glacier retreated for about 125 years). 

Finally, 2990 m is the glacier forehead, 2844–2950 m [1] is the succession stage for Populus purdomii forest, which acts as the broadleaf deciduous stage, 2800–2970 m (the present study) is the whole succession stage from pioneer plants to climax communities. Because of the different chronosequences, the lat/long coordinates and altitude of Yang et al. [1]and the present study are also different. 

4.Were the biomass estimates separate estimates from that of Yang et al. (2014)? If so, some comparison with those biomass and N accumulation rates should be presented in the Results and Discussion.

Reply: The biomass estimates were different from the reports of Yang et al. [1], but the same as those from Yang et al. [2] Specific biomass estimation methods and related research results have been explained above. We have also included comparisons of biomass and N accumulation rates in the Results and Discussion sections, and new contents are highlighted in red in the revised manuscript, such as in lines 245-250.

5.The regression statistics used (line 161) do not seem like they are appropriate for the statements made about the slope being different from one. Without the data, I can only do a visual estimate, but the significance level of p<0.001 seems inappropriate for the presented data, especially for tree, herb and the A layer. A more detailed statistical explanation is needed.

Reply: All statistical analyses were conducted using SPSS 21.0. The relationships between the relative change rates of N and P were analyzed by linear regression. A more detailed statistical explanation was added to the supplementary material.

RESULTS and DISCUSSION

1.Many factors limit vegetation growth, both individually and in concert with each other. Moisture, sunlight, temperature, nutrients, etc. can result in the observed rate of biomass accumulation. The reported rates of biomass accumulation across this chronosequence seem comparable with other estimates of forest growth rates in similar climates. These glacial ecosystems do not appear to be growth-limited in any unusual ways. Accepting that this study seeks only to examine N and P limitations, it is a fair question to wonder about N vs. P limitations. However, I think it is useful for the authors to present this context of many limitations, and their particular interest in N and P, to their readers.

Reply: In the Hailuogou glacier retreat area, which is about 2000 m in length and 200 m in width, it is generally believed that the moisture, sunlight, and temperature control the plant belt spectrum and forest line formation changes little, which cannot be the main reason for the control of the emergence and succession of vegetation in bare land. Therefore, the nutrient elements necessary for plant growth may be the main reason for the development of vegetation. In this manuscript, the limitation of N and P in this succession sequence was examined through the distribution and accumulation characteristics of N and P, and the N:P ratio. The overriding goal was to better understand nutrient controls on ecosystem development.

2.As an alternative interpretation to their conclusion of N limitation, I might suggest that the accumulation of N in the soil demonstrates the build up of "extra" N. "Tight" cycling of N has been demonstrated in many forested ecosystems, such that the pool of organic N in the forest floor and the soils remains relatively constant over successional time. The absence of this "tight" cycling of N in this primary succession could be interpreted as an indication of the relative availability of N. P on the other hand, is much more stable over the chronosequence in the litter, O and A layers, possibly suggesting that the continuing requirement of P by accumulating biomass takes up any new P from decomposing organic matter while accessing the continuing release of P from primary (and possibly secondary) minerals to fine roots and mycorrhizae.

Reply: The reviewer's interpretation provides another way to explain the dynamic changes in N and P from the source and cycle of N and P. The N concentration in the original bare land of Hailuogou was almost 0 [2]. Through biological N fixation, atmospheric N deposition, and litter decomposition, the N in the surface soil is continuously accumulated, and the nutrient conditions are improved. Because the glacial retreat of Hailuogou is only around 120 years old, it is still at a "young stage" relative to the soil age, and no complete soil profile has been formed. Although the succession of forests has formed the climax community, soil development continues. Therefore, N in the soil may continue to accumulate.

In this succession sequence, forest succession and the dynamics of soil P are an interactive process, especially after the formation of a coniferous forest climax community dominated by A. fabri and P. brachytyla. On the one hand, these coniferous forest trees have a stronger ability to secrete organic acids; on the other hand, the acidity of needle leaf litter is lower, which makes the soil pH significantly lower in the sampling site of glacier retreat for 125 years (pH = 4.4). However, the rapid decrease in soil pH results in the accelerated release rate of primary mineral P, and the Pmineralization pool ( the supply of available P from mineralization of organic P in the organic layer) was ~2.9 times that of the P requirement [7]. This is consistent with the reviewer's explanation.

3.In the Discussion the authors state: "Generally, a N:P ratio <14 in leaves indicates that plant growth is limited by N, whereas a N:P ratio >16 indicates that plant growth is limited by P." (line 307) So it is curious that they do not report their own N:P ratio in leaves as a comparison. Total vegetation N:P ratios are a weaker absolute indicator of nutrient limitations due to the various N:P ratios in different species and different vegetation tissues. They cite (line 311) a reported range from "<10 for N limitation" to ">20 for P" across a broad range of vegetation. Their reported ratio is just at the boundary of 10, which does not make a strong case for N limitation.

Reply: In the corresponding part of the manuscript, I added content related to the N:P ratio of leaves on each dominant tree species and the whole tree layer. The results are as follows:

Table 1. N: P ratio of leaves in different species and tree layers

 N:P ratio in leaves

Sampling site Glacier retreat year H. rhamnoides Salix spp. P. purdomii A. fabri Tree

S1 15y 19.63 4.16 13.68 10.99

S2 35y 18.16 5.00 15.42 10.14

S3 45y 17.79 10.14 14.81 10.45 14.30

S4 57y 14.89 8.06 12.84 10.01 12.68

S5 85y 12.61 10.37 10.42

S6 125y 11.25 11.25

It can be seen from the Table 1 that the N:P ratios of different dominant trees were quite different. As a N-fixing plant, the N concentration in H. rhamnoides leaves was high, so the N:P ratio in its leaf was higher than that in other dominant tree species, whereas the mean P concentration in Salix spp. leaves was 4.17 g kg-1, which was higher than that of the other tree species. Therefore, the N:P ratio in its leaves was extremely low. Due to the different absorption of N and P by different tree species, the concentration of N and P in leaves of different tree species were different. Therefore, when we consider the N: P ratio of vegetation leaves in each sample plot, we consider the N: P ratio of tree layer leaves after weighting. From S1 to S6 sites, the N:P ratio in tree leaves ranged from 10.14 to 14.30, except in the S3 site, and the N: P ratio of other sites was significantly lower than 14 (N:P ratio <14 in leaves indicates that plant growth is limited by N). In addition, the N: P ratio of leaves of A. fabri, for which the biomass of the ecosystem at the S6 site was >89% [2], was also found to be lower than 14. Therefore, only from the N:P ratio of leaves did we tend to consider that the growth of plants was probably limited by N. 

We mainly studied the dynamics changes in N and P pools in the whole primary succession and discussed the relationship between N and P in the ecosystem. Therefore, when we study N:P stoichiometry of ecosystems, we should consider carefully the pools of N and P [8]. Thus, a given N:P ratio in the present study may refer to total N and P pools, and this can reflect the N: P ratio of the whole vegetation layer. Thus, we describe a reported range from <10 for N limitation to >20 for P limitation across a broad range of vegetation. The N:P ratio for vegetation ranged from 8.35 to 13.25 (S1: 13.25, S2: 9.66, S3: 9.79, S4: 9.01, S5: 8.35, S6: 10.00) and remained constant after the S2 site.

The N:P ratio in different study areas, ecosystems, and vegetation types might vary greatly. Therefore, most scholars accept the view of ecosystems being N-limited at a low N:P ratio and P-limited at a high N:P ratio. Although the N:P ratio of leaves and vegetation layer are different to some extent, these ratio were lower than the values currently thought to reflect N limitation. The present study sought only to examine N and P limitations through their N:P ratio.

It is indeed difficult to assess N or P limitation to plants by only relying on stoichiometry ratios of N and P. We need more evidence to support this conclusion. Therefore, combined with previous studies, I presented more evidence, including soil nutrient supply and nutrient-addition experiments, to support that the growth of plants in the Hailuogou glacier retreat area may limited by N. First, Zhou et al. [9-11]carried out much research on soil P in the Hailuogou glacier chronosequence. Their results showed that: at 120-y-old-site, the Pavailable pool in the organic layer and 0–6 cm mineral soils was 27.0 kg ha−1 and ~5.3 times the annual plant P requirement; the Pmineralization pool, representing the supply of available P from mineralization of organic P in the organic layer, was ~2.9 times the P requirement; the P weathering pool was 7.5 kg hm−2 y−1 and was higher than the P requirement. These results suggest that the current P pools can offer enough P for the growth of the ecosystem. Second, although it is difficult to carry out a nutrient-addition experiments in this primary succession sequence, a N-addition experiment on A. fabri seedings may be helpful to understand nutrient controls on plant growth. Yang et al.[12]conducted such an experiment in the Gongga Mountain observation station, which is ~1 km away from the Hailuogou glacier retreat area. They found that the total biomass, leaf dry weigh, leaf mass ratio, leaf N and P concentration, and leaf N:P ratio of the A. fabri seedlings increased by 11.29%, 46.70%, 41.40%, 37.30%, 22.33%, and 6.43%, respectively, after 2 years addition of N (50 kg N hm−2 y−1), indicating that the growth of A. fabri seedlings was probably limited by N. Furthermore, our study also showed that the biomass accumulation rate may be more positively correlated with the accumulation rate of N. Therefore, from the results of N:P stoichiometric of vegetation, soil P supply, and N-addition experiments, we suggest that the plant growth in the Hailuogou glacier succession sequence may be limited by N.

The text has been revised to address this in lines 257-265, and 387-407.

4.Line 41 of the Abstract states: "... N was the main limiting factor for plant growth in this sequence." Given that an abstract may be the only part of the research that some scientists may read, I would suggest that this conclusion needs to be stated more carefully, perhaps with reservations.

Reply: We have modified the corresponding statements in the article and the amendments are highlighted in red in the revised manuscript, such as lines in 41, 104, 376, 384, 406, 421. 

SPECIFIC COMMENTS

line 98: If an actual average can not be calculated from recorded data, then two to four significant figures are not needed to give the reader an approximate estimate for temperature and rainfall. In this case ~4 Deg C and ~2000 mm rain would suffice.

Reply: We have revised the corresponding part of the paper to make the expression more scientific and the amendments are highlighted in red in the revised manuscript, such as lines in 110.

line 120: all of the numbers in Table 1 should include at most two significant figures. The estimates are not more precise than that, and having fewer digits to examine makes it easier for the reader.

Reply: We have revised the numbers in Table 1 to clarify the results for the reader.

Table 1. Investigation of vegetation and soil at different glacier retreat times 

Sample site S1 S2 S3 S4 S5 S6

Glacier retreat year 15 y 35 y 45 y 57 y 85 y 125 y

Dominant trees H. rhamnoides,

Salix spp.,

P. purdomii H. rhamnoides,

Salix spp.,

P. purdomii P. purdomii (half-mature),

A. fabri P. purdomii (mature),

A. fabri A. fabri,

P. brachytyla

 A. fabri,

P. brachytyla

Biomass

 (t hm−2) 7 (1) 120 (14) 198 (23) 224 (10) 281 (24) 375 (29)

Surface soil pH 6.9 (0.3) 6.4 (0.5) 5.5 (0.5) 5.6 (0.4) 5.2 (0.4) 4.4 (0.4)

Thickness

(cm) O layer 0.8 (0.1) 1.1 (0.2) 1.4 (0.23) 1.8 (0.3) 2.3 (0.9) 3.6 (0.8)

 A layer -- 1.8 (0.7) 2.6 (0.2) 3.8 (0.9) 4.6 (1.2) 5.4 (1.2)

Bulk density

(g cm−3) O layer 0.13 (0.01) 0.12 (0.02) 0.11 (0.06) 0.12 (0.07) 0.19 (0.03) 0.35 (0.05)

 A layer -- 0.31 (0.11) 0.38 (0.10) 0.33 (0.09) 0.25 (0.06) 0.30 (0.09)

Data shown as means with standard deviation in parentheses.

line 121: Tree biomass was estimated using the allometric equations reported by Liu (2019). Are these the same equations reported by: X Zhong, N Wu, J Luo, K Yin, Y Tang, Z Pan - Chengdu Science and Technology …, 1997. Researches of the forest ecosystems on Gongga Mountains.

Reply: They are not the same equations. Liu et al.[3] estimates were based on the measured biomass of the main tree species in the primary forest succession in the Hailuogou glacier retreat area, the total biomass of the trees and the biomass of different components (such as branches, leaves, trunks and roots) together with the breast diameter and tree height. The allometric equations were for four common tree species: A. fabri, P. purdomii, H. rhamnoides, and Salix spp.

line 125: The difference between "herbs" and "ground cover" is not clear.

Reply: "Herbs" are herbaceous plants, while "ground cover" refers to moss. We revised the "ground cover" to "moss" in the manuscript to clarify this point. We revised this for clarity in lines 141, 211, 255, 238, 246.

line 143: "The thickness and bulk density of each soil layer were then measured using a measuring tape and a cylindrical tube, respectively." Soil is notoriously hard to sample because of the inherent spatial variation. It would be helpful to include the N (sample size at each site). Table 1 also needs to indicate the unit of the variation (one standard deviation, one standard error (standard deviation of the mean), two

standard errors?). Pit sampling (TG Huntington, DF Ryan, et al. 1988 Estimating soil nitrogen and carbon pools in a northern hardwood forest ecosystem. in Soil Science Society of Am.) provides a more accurate bulk density and nutrient concentration estimate. For ease of examining the data, it would be very helpful to use two/three significant figures, especially given the high variation that seems apparent, e.g. 120.80 +- 13.89 should probably be reported as 120 +- 14; 6.43 +-0.53 is probably best as 6.4 +- 0.5.

Reply: We have revised the data in Table 1 to include data about biomass, surface soil pH, thickness, and bulk density, including the mean value and one standard deviation.

line 174: "vegetation N pool sharply increased" The word "sharply" does not seem appropriate here as the accumulation rate of N over the 125 years appears linear. "Sharply" is used again on line 229.

Reply: We have removed the word "sharply" regarding the vegetation N pool, such as lines in 207 and 291.

line 201: The rates of relative N and P changes in Figure 3 are not consistent with the explained methods. Equation (1) line 157 shows the calculation of rate of relative change for each site (i = 1– 6). Figure 3 shows more than 6 points. The construction of Figure 3 needs to be explained.

Reply: Figure 3 shows the line regression about the relative change rates of N and P. We divided the ecosystem into six components: tree, shrub, herb, moss, O layer, and A layer. For each component, the rate of relative N or P change was calculated as follows: the relative N or P changes were calculated as the N or P pool at the current age stage divided by that at a previous age stage. Then the rate of relative N or P changes was obtained by the relative N or P pool change divided by the age interval between two adjacent age stages [1, 13].For example, rate of relative N or P change in trees:

where Rtree is the rate of relative N or P change in trees; S represents the tree N or P pool; T represents the time of glacier retreat; and i is each site (i = 1–6). Each site had three replicates. More detailed information was added to the supplementary material.

line 202: The statistics associated with Figure 3 indicate a significant correlation between N and P (i.e. a slope significantly different than 0). However, in order to make a statement about the importance of the slope being different than 1.0, the 95% confidence bounds on the slope estimate (e.g. slope of 1.26 for trees) needs to be provided. Also Figure 3 shows a slope of 1.26while line 206 states a slope of 1.32. Other inconsistencies between the text and Figure 3 are also presented.

Reply: We have revised the mistakes pointed out by the reviewer. The 95% confidence bounds on the slope were: tree: 1.265 ± 0.245, shrub: 0.848 ± 0.066, herb: 0.874 ± 0.323, moss: 0.744 ± 0.264, O layer: 0.735 ± 0.343, and A layer: 0.873 ± 0.207. We revised the text to address this point in line 241.

line 299: "also showed that the rate of relative N accumulation was faster than that of P in surface soil." Without knowing if the reported slopes were significant or not (see comment above, line 202), the results of this study may only "suggest." However, the point here should probably be about which processes are causal and which observations are just incidental to those processes. A higher relative N accumulation rate WILL result in a change in N:P ratio, in all cases. The more salient question is how N and P cycling are changing (relative to each other) as the forest stand matures.

Reply: The slope of regression for the rate of relative N and P changes was compared with 1, and we were able to detect how the accumulation of N changed with P accumulation along the successional gradients. Thus, we can understand the relationship between N and P for different ecosystem components to clearly understand N–P interactions along the successional gradients. The threshold p value of regression in the O layer and A layer was = 0.001 and <0.001, respectively, indicating that the results were significant. However, making a causal claim without knowing which processes are causal and which observations are incidental to those processes is not appropriate. Therefore, We have revised the manuscript accordingly with more appropriate interpretation or hypotheses. In the corresponding part of the paper, We revised “showed” to “suggested”.

 The more salient question raised by expert is the focus of our next research. Examining the nutrient cycling is to understand the N and P utilization, circulation, and transmission among different components of the ecosystem. Especially in different succession stages, the different dominant species have different ways of nutrient cycling, which will lead to differences in N, P concentration and N:P ratio.

line 301: In this primary successional sequence it seems unlikely that P has weathered out of the rocks and is less available at the end of the chronosequence.

Reply: Zhou et al. [11]reported that the weathering processes in the Hailuogou glacier chronosequence are rapid due to fast vegetation succession, higher temperatures, and relatively high precipitation. In addition, Zhou et al. [9] reported the average rate of weathering of primary mineral phosphate (RLP) in this chronosequence. The average RLP (14.1 mmol m−2 y−1) in the Hailuogou glacier chronosequence was ~47 times higher than the global rate of P release. Especially at the 120-y-old site, the RLP was significantly higher than at sites with similar ages in temperate and subtropical zones. Zhou et al. [11] reported the changes in soil P speciation along this chronosequence and indicated that the concentration of bioavailable P in surface soil showed a trend of increasing, with 5–11.5% of total soil P, and the stocks of bioavailable P were greater than the annual P requirement of vegetation. In this primary successional sequence, P had not only weathered out of the rocks but was also available at the end of the chronosequence.

line 316: "owing to the high N level of N-fixing" adding "perhaps due to" would be a fairer statement of likelihood. This study did not establish the biogeochemical role of N-fixing microbes and plants at these sites. Making a causal claim about what particular biogeochemical process alters the relative uptake of nutrients is therefore not appropriate as a statement, and more appropriate as an interpretation or hypothesis.

Reply: Leguminous plants (such as Astragalus adsurgens Pall. and Astragalus souliei Simps) and H. rhamnoides are the dominant species in the S1 site. As N-fixing plants, their N concentration was higher than that of other species, which may lead to the higher N:P ratios. However, we did not establish the biogeochemical role of N-fixing microbes and plants at this site, so making a causal claim is not appropriate. We revised “owing to” to “ perhaps due to”, such as line in 379.

line 326: "This increase in N and decrease in P may shift the factor limiting plant growth from N to P after several hundred years. Therefore, the dynamics of the N:P ratio in this forest primary succession needs further study." Successional changes in relative nutrient availability are important to document, and this 125 year sequence provides a good example of nutrient dynamics in a chronosequence. However, the N limitation assertion is an hypothesis, and the authors should be careful in how they state their interpretations. Given the various ratios of N:P they report at their sites, the N limitation is possible, but not conclusive.

Reply: We have revised the manuscript according to this comment to offer interpretation or hypothesis statements, which are more appropriate, such as lines in 41, 104, 376, 384, 406, 421. 

line 339: "whereas the N:P ratio in vegetation maintained a constant low level due to the tree layer having a rapid P accumulation rate compared with N;" "higher" rather than "rapid" is probably more appropriate here. Plants and ecosystem accumulate nutrients according to their needs, and we can only infer limitation if we do not determine this experimentally (e.g. with a nutrient addition experiment).

Reply: In the corresponding part of the manuscript, We revised “rapid ” to “ higher”, such as lines in 240, 241, 350, 358, 382.

Reference

1.Yang Y, Wang GX, Shen HH, Yang Y, Cui HJ, Liu Q. Dynamics of carbon and nitrogen accumulation and C: N stoichiometry in a deciduous broadleaf forest of deglaciated terrain in the eastern Tibetan Plateau. For. Ecol. Manag. 2014; 312:10-18.

2.Yang DD, Luo J, She J, Tang RG. Dynamics of Vegetation Biomass Along the Chronosequence in Hailuogou Glacier Retreated Area, Mt.Gongga. Ecology and Environmental Sciences, 2015; 24(11):1843-1850.

3.Liu T, Shang HL, Luo J, Sun SQ, He QM, Li AD, Zhang J. Allometric equations for four common tree species in retreated area of Hailuogou Glacier, Gongga Mountain. Southwest China J. Agric. Sci., 2019, 32:922-928.

4.Wang X, Yuan W, Feng XB, Wang DY, Luo J. Moss facilitating mercury, lead and cadmium enhanced accumulation in organic soils over glacier erratic at Mt. Gongga, China. Environment Pollution, 2019,254: 112974.

5.Wang X, Luo J, Lin CJ, Wang DY, Yuan W. Elevated cadmium pollution since 1980s recorded by forest chronosequence in deglaciated region of Gongga, China. Environment Pollution, 2020a, 260:114082.

6.Wang X, Luo J, Yuan W, Lin CJ, Wang FY, Liu CL, et al. Global warming accelerates uptake of atmospheric mercury in regions experiencing glacier retreat. Proceedings of the National Academy of Sciences of the United States of America, 2020b, 117(4): 2049-2055.

7.Zhou J, Sun HY, Wang JP, He QQ, Bing HJ, Wu YH. Comments on “unravelling on community assemblages through multi-element stoichiometry in plant leaves and roots across primary uccessional stages in a glacier retreat area” by Jiang et al. Plant Soil, 2018, 433: 1-5.

8.Zhang LX, Bai YF, Han XG. Application of N:P stoichiometry to ecology studies. Acta Bot. Sinica, 2003; 45(9):1009-1018.

9.Zhou J, Bing HJ, Wu YH, Sun H, Wang J.Weathering of primary mineral phosphate in the early stages of ecosystem development in the Hailuogou Glacier foreland chronosequence, European Journal of Soil Science, 2018,69:450-461.

10.Zhou J, Bing HJ, Wu YH, Yang ZJ, Wang JP, Sun HY, et al. Rapid weathering processes of a 120-year-old chronosequence in the Hailuogou Glacier foreland, Mt. Gongga, SW China. Geoderma, 2016; 267:78-91.

11.Zhou J, Wu YH, Prietzel Jörg, Bing HJ, Yu D, Sun SQ, et al. Change of soil phosphorus speciation along a 120-year soil chronosequence in the Hailuogou Glacier Retreat Area (Gongga Mountain, SW China). Geoderma, 2013, 195-196(3):251-259.

12.Yang L, Wang G, Yang Y, Yang Y (2012) Responses of leaf functional traits and nitrogen and phosphorus stoichiometry in Abies fabri seedlings in Gongga Mountain to simulated nitrogen deposition. Chinese Journal of Ecology 31:44–50 (in Chinese with English abstract).

13.Yang YH, Luo YQ, Finzi AC. Carbon and nitrogen dynamics during forest stand development:a global synthesis. New Phytol, 2011; 190: 977-989.

Replies to reviewer 3

Reviewer #3: The manuscript entitled “Dynamics of nitrogen and phosphorus accumulation and their stoichiometry along a chronosequence of succession forests in the Hailuogou Glacier retreat area, eastern Tibetan Plateau” focused on exploring the N and P accumulation dynamics and their stoichiometry during forest primary succession in a glacier retreat area on the Tibetan Plateau. However, they didn’t address any the following key scientific issues that they raised. 

1.I do not agree with you that surface soil N increased with increasing years of glacier retreat, becoming a main N pool, whereas increased P accumulation in vegetation after 125 y of recession indicated that much of the soil P was transformed into the biomass P pool. Obviously, the increasing N:P ratio for surface soil do not support the conclusion of N or P pool. You should provide the solid evidence. 

Reply: Accumulation of N in forests has received much attention in previous studies. However, soil N pools either increased [1,2] or decreased [3,4] with forest succession development. In the Hailuogou glacier retreat area, the N concentration in moraine is extremely low, and the available N is almost 0. However, with forest succession and soil development, the N in soil increased significantly. After 125 y of glacier retreat, the N pool in surface soil is 4859.60 kg hm−2, and surface soil N comprised 68% of the total ecosystem N, and only 32% was stored in vegetation. Furthermore, the N accumulation was also detected in surface soil at 41 kg hm−2 y−1 is this chronosequence, which was higher than that in vegetation (17.99 kg hm−2 y−1). Additionally, after 15 y of glacier retreat, 97% of the total ecosystem N was stored in vegetation, and only 3% of this amount was found in surface soil. However, after 125 y of glacier retreat, surface soil N comprised 68% of the total ecosystem N, and only 32% was stored in vegetation. According to these results, we suggest that N is mainly stored in soil, and the soil pool may the main sink for atmospheric N [5]. 

Weathering of minerals from parent rock material is the principal source of P in terrestrial ecosystems [6,8]. Zhou et al. [9–11] researched the weathering of primary mineral P and soil P in the Hailuogou chronosequence. They reported that the concentration of bioavailable P in surface soil showed a trend of increasing with percentages of 5–11.5% of total soil P, and the total P pool bound in tree biomass was about 75% of the soil TP stock in the A layer in the late stage, whereas it only occupied 3.8% of the TP stock in the C layer. These results might indicate that the P pool in surface soil is larger and more easily transported to the biomass P pool than the C layer. In our study, we researched the total ecosystem P pool along this forest primary succession, including vegetation and surface soil. The P pool in vegetation increased from 7.06 kg hm−2 to 232.48 kg hm−2, and a major change in P distribution was also detected in vegetation. After 15 y of glacier retreat, only 28% of the total ecosystem P was stored in vegetation, whereas after 85 y of glacier retreat, 63% of the total ecosystem P was found in vegetation, and the P pool in vegetation was equivalent to that in surface soil after 125 y of glacier retreat. According to the change in P distribution in vegetation and surface soil, we might also suggest that much of the soil P was transformed into the biomass P pool at the end of this succession sequence.

Through comparing the regression slopes for the rate of relative N or P changes with 1 in the 95% confidence intervals, we were able to detect how the accumulation of N changed with P accumulation and understand the relationship between N and P for different ecosystem components to clearly understand N–P interactions along this succession sequence. In the O and A layers, the slope between the rates of relative N and P changes was 0.73 and 0.87 (<1), respectively, which indicated that the rate of relative N change in surface soil was higher than that of relative P change. The present study showed a significantly increased N:P ratio in surface soil with increasing glacier retreat years. The increased N:P ratio could be attributed to a more higher rate of relative N accumulation compared with the rate of relative P accumulation in surface soil. This N-P stoichiometric relationship can also be reflected in the distribution of N and P pools for surface soil in the ecosystem. The N accumulation rate of surface soil was higher, and the proportion of N storage in the ecosystem increased from 3% to 68%, while the rate of P accumulation in surface soil was relatively lower, and the proportion of P storage in the ecosystem decreased from 72% to 36%. The increase in the proportion of N and the decrease in P in surface soil can also lead to the increase in the N: P ratio in surface soil. 

2.some fitting results are incredible. For example, the fitting of N:P ratio for vegetation is straight line? The results of this study may be false positive. As a result, I doubt scientific values and the conclusions in this study. 

Reply: The data in this study are authentic and reliable, and all relevant data involved in the calculations are provided in the supplementary materials. We used the logarithmic model to fit the relationship between the N:P ratios for vegetation and glacier retreat year. However, the results showed P > 0.05, indicating that the logarithmic model had no statistical significance. Therefore, the fitting of the N:P ratio for vegetation was not a straight line. We have revised and explained the corresponding part of the manuscript, such as lines in 271-274.

3.the MS was poorly written in the language and logic, which is needed to be substantially strengthened before submission. The production of the tables and figures are also very unprofessional. For example, Table 2, I can not understand. The authors do need to pay attention to these issues for any scientific article. 

Reply: We modified Table 2 to clarify the results for the reader. Furthermore, the manuscript has been polished by the professional polishing company Editage. We have substantially strengthened the language and logic to make it more rigorous and scientific.

Table 2. N and P pools in various ecosystem components

Sampling site S1 S2 S3 S4 S5 S6

Glacier retreat year 15 y 35 y 45 y 57 y 79 y 125 y

Tree N pool

(kg hm−2) 75.3 (10.4)e 701.5 (99.6)d 790.5 (103.2)cd 937.6 (69.7)c 1380.9 (95.4)b 1863.7 (122.9)a

 P pool

(kg hm−2) 6.3 (1.5)e 73.9 (5.5)d 83.6 (8.2)d 106.8 (3.5)c 183.2 (13.2)b 199.1 (10.6)a

Shrub N pool

(kg hm−2) 0.3 (0.1)d 0.6 (0.1)d 31.6 (2.3)cd 74.9 (17.8)c 148.1 (21.4)b 319.1 (58.4)a

 P pool

(kg hm−2) 0.01 (0.0)c 0.1 (0.0)c 3.2 (0.2)c 7.7 (1.8)b 10.7 (1.5)b 22.4 (4.1)a

Herb N pool

(kg hm−2) 4.4 (0.1)b 7.6 (1.6)b 7.1 (0.9)b 23.5 (5.1)a 7.7 (1.7)b 22.6 (6.6)a

 P pool

(kg hm−2) 0.2 (0.0)d 0.9 (0.2)c 1.1 (0.1)c 3.5 (0.6)a 0.6 (0.1)c 2.2 (0.2)b

Moss N pool

(kg hm−2) 1.2 (0.3)d 7.0 (1.9)cd 22.8 (1.6)c 18.6 (2.4) cd 150.7 (23.4)a 85.3 (6.1)b

 P pool

(kg hm−2) 0.1 (0.0)d 0.9 (0.3)cd 1.7 (0.3)c 1.7 (0.4)c 11.2 (1.7)a 7.4 (0.9)b

Litter N pool

(kg hm−2) 12.6 (0.4)d 31.7 (2.1)c 40.7 (0.9)a 42.6 (0.9)a 42.2 (0.9)a 35.4 (0.9)b

 P pool

(kg/hm2) 0.5 (0.0)c 1.4 (0.1)b 1.5 (0.1) b 1.9 (0.1)a 1.3 (0.1) b 1.4 (0.1)b

O layer N pool

(kg hm−2) 3.7 (0.1)d 273.4 (23.4) c 255.0 (10.9)c 433.8 (61.5)b 502.5 (68.6)b 1919.2 (231.2)a

 P pool

(kg hm−2) 17.7 (3.7)c 18.7 (3.8)c 20.3 (6.3)c 26.8 (4.5)bc 30.5 (5.9)b 126.1 (2.6)a

A layer N pool

(kg hm−2) -- 1085 (59.2)d 1581 (355.7)c 2065 (297.1)b 2184.7(110.6)b 2940.5 (34.7)a

 P pool

(kg hm−2) -- 83.8 (13.8)b 103.2 (28.0)ab 129.7 (12.3)a 87.1 (6.7)b 130.3 (7.4)a

Data shown as means with standard deviation in parentheses. Different letters in the same row indicate significant difference among N or P pools among different sites.

Reference

1.Mao R, Zeng DH, Hu YL, Li LJ, Yang D. Soil organic carbon and nitrogen stocks in an age-sequence of poplar stands planted on marginal agricultural land in Northeast China. Plant Soil. 2010; 332: 277–287

2.Adam F.A., Pellegrni, Willam A. Hoffmann. Carbon accumulation and nitrogen pool recovery during transitions from savanna to forest in central Brazil. Ecology, 2014, 95(2):342-352.

3.Pang XY, Hu H, Qiao YK, Pan KW, Liu SQ, Chen QH, et al. Nutrient Distribution and Cycling of Artificial and Natural Subalpine Spruce Forests in Western Sichuan. Chin J Appl Environ, 2002, 8(1):1~7.

4.Hooker, T.D., Compton, J.E. Forest ecosystem carbon and nitrogen accumulation during the first century after agricultural abandonment. Ecol. Appl. 2003; 13 (2); 299–313.

5.Aber J, McDowell W, Nadelhoffer K, Magill A, Berntson G, Kamakea M, et al. Nitrogen saturation in temperate forest ecosystems: hypothesis revisited. Bioscience, 1998, 48:921–934.

6.Filippelli, G.M. The global phosphorus cycle: past, present, and future. Elements, 2008; 4 (2): 89–95

7.Wu H, Zhou J, Yu D, Sun SQ, Luo J, Bing HJ, et al. Phosphorus biogeochemical cycle research in mountainous ecosystem. J Mount Sci, 2014; 10: 43–53.

8.Schlesinger W.H, Bernhardt E.S. Biogeochemistry-an Analysis of Global Change. Elsevier Publishers, The Kingdom of the Netherlands, 2016.

9.Zhou J, Bing HJ, Wu YH, Sun H, Wang J.Weathering of primary mineral phosphate in the early stages of ecosystem development in the Hailuogou Glacier foreland chronosequence, European Journal of Soil Science, 2018,69:450-461.

10.Zhou J, Bing HJ, Wu YH, Yang ZJ, Wang JP, Sun HY, et al. Rapid weathering processes of a 120-year-old chronosequence in the Hailuogou Glacier foreland, Mt. Gongga, SW China. Geoderma, 2016; 267:78-91.

11.Zhou J, Wu YH, Prietzel Jörg, Bing HJ, Yu D, Sun SQ, et al. Change of soil phosphorus speciation along a 120-year soil chronosequence in the Hailuogou Glacier Retreat Area (Gongga Mountain, SW China). Geoderma, 2013, 195-196(3):251-259.

---

## [Decision Letter · Decision Letter 1]

29 Dec 2020

PONE-D-20-29430R1

Dynamics of nitrogen and phosphorus accumulation and their stoichiometry along a chronosequence of forest primary succession in the Hailuogou Glacier retreat area, eastern Tibetan Plateau

PLOS ONE

Dear Dr. Peng,

Thank you for submitting your manuscript to PLOS ONE. After careful consideration, we feel that it has merit but does not fully meet PLOS ONE’s publication criteria as it currently stands. Therefore, we invite you to submit a revised version of the manuscript that addresses the points raised during the review process.

Please see attached review comments for details. 

We look forward to receiving your revised manuscript.

Kind regards,

Dafeng Hui, Ph.D.

Academic Editor

PLOS ONE

Additional Editor Comments (if provided):

The reviewer has some additional minor concerns. Please address them in a revision.

Reviewers' comments:

Reviewer's Responses to Questions

**Comments to the Author**

1. If the authors have adequately addressed your comments raised in a previous round of review and you feel that this manuscript is now acceptable for publication, you may indicate that here to bypass the “Comments to the Author” section, enter your conflict of interest statement in the “Confidential to Editor” section, and submit your "Accept" recommendation.

Reviewer #1: (No Response)

2. Is the manuscript technically sound, and do the data support the conclusions?

Reviewer #1: Partly

3. Has the statistical analysis been performed appropriately and rigorously? 

Reviewer #1: No

4. Have the authors made all data underlying the findings in their manuscript fully available?

Reviewer #1: Yes

5. Is the manuscript presented in an intelligible fashion and written in standard English?

Reviewer #1: Yes

6. Review Comments to the Author

Reviewer #1: (No Response)

7. PLOS authors have the option to publish the peer review history of their article (what does this mean?). If published, this will include your full peer review and any attached files.

Reviewer #1: **Yes: **Deane Wang

---

## [Author Response · Author response to Decision Letter 1]

13 Jan 2021

Overall Comments for Editor and Authors

The manuscript is much improved with the changes the authors have made. However, as I learned more about the research site and now had access to the Supporting Information sections, a few additional issues came up. These are not insurmountable, but I believe are important issues that the authors should attend to. I continue to think that the paper contains an important data set and provides new ecosystem-level information about this important research site.

For the Hailuogou Glacier foreland, the on-going debate in the literature about N versus P limitations is an essential part of the Introduction. Your analysis reflects the complete ecosystem N:P story for the entire chronosequence, which is apparently the first place this would be published, and is therefore important to document. It is important to clearly layout the sequence of ideas and appropriate publications that describe the rationale for proposing that N is an important limiting factor (as opposed to P), with your broader assertion that this is true across the entire chronosequence. It should not be the job of the reader to piece together this rather complicated history of publications on N and P in vegetation and soils in this important glacial research area. Four examples of places where the Introduction and Discussion could clarify the relationship of the current research (your paper) and past publications are listed below. There may be other places to provide a more complete context as well.

For example:

1) The debate over N versus P limitations initiated in Jiang et al. 2018 and followed up by Zhou et al. 2018, are not mentioned in the Introduction. Why? The Zhou commentary includes estimates of available P in the O and A horizons in the 120 year-old sequence, clearly relevant to the data and interpretations that you present.

Zhou et al. also state:

“ Moreover, as has been discussed in the literature, it is hard to assess P limitation to plants only relying on stoichiometry ratios between N and P (e.g. Craine et al. 2008; Güsewell 2004; Sullivan et al. 2014). For example, Güsewell (2004) argued that low N:P ratios represent N limitation, but a consistent interpretation of intermediate and high N:P ratios is not achieved, suggesting that the N:P ratios cannot be the only criterion for assessing nutrient limitation.”

This is a key contention that you should directly address, perhaps both in your Introduction and your Discussion.

Reply: Following the suggestions made by the Reviewers we have revised this section. Please check lines 346-389.

2) Yang et al. 2015 presents biomass data across the chronosequence that you note are the same data set that you employ to do the N:P analyses. The biomass data are very similar but not identical. If this builds upon the same research effort, then it should be stated that this research is an extension of that work (for example: "biomass data are from Yang et al. 2015"). The reported data should be identical.

Biomass in Table 1: 7 (1) 120 (14) 198 (23) 224 (10) 281 (24) 375 (29)

From Abstract Yang 2015: total living: 10.195 Mg·hm~(-2) to 366.122 Mg·hm~(-2), from tree alone: 9.162 Mg·hm~(-2) to 332.461 Mg·hm~(-2).

Yang DD, Luo J, She J, Tang RG. Dynamics of Vegetation Biomass Along the Chronosequence in Hailuogou Glacier Retreated Area, Mt.Gongga. Ecology and Environmental Sciences, 2015; 24(11):1843-1850.

Reply: In their work, Yang et al. 2015 report the total living biomass and tree biomass whereas the biomass expressed in Table 1 of the present paper is the total biomass of the ecosystem, including both living vegetation and litter. Therefore, biomass differs between the studies. Additionally, site S1 in Yang et al. corresponded to 17 y of glacier retreat, while in our study it corresponded to 15 y of glacier retreat. As so, the biomass in our S1 was a little lower than that in Yang et al. S1. However, as the same dataset is used in both papers, and thus should be identical, we have modified the information in Table 1.

Additionally, we also made some modifications in the text referring to the collection of plant samples. Pleased check lines 148-149.

3) On Line 99 you state: “However, few researchers have estimated the vegetation P pool size, the characteristics of N and P accumulation, and the N: P ratio in this successional sequence.”

You cite Jiang et al. 2018, but only with respect to soil layers (line 160) and P requirements

Line 362: The rate of relative P change was higher than the relative N change in the tree layer, especially with the greater requirements for P of A. fabri and P. brachytyla [57], which led the N:P ratio being maintained at a lower level.

The Jiang paper clearly examines N:P stoichiometry at the same site. This should be in your Introduction (literature review).

Reply: Following your suggestion, we have added information to the manuscript that illustrates this point. Please check lines 98-105 in the Introduction section and lines 346-366 in the Discussion section.

4) Jiang et al. estimate total above-ground biomass: "Total above-ground biomass increased abruptly in the beginning five stages, and then increased slightly in stages 6 and 7 (their Fig. S4 a)." If your values contrast these estimates, then you need to explain. Your two studies could be complementary or competing. The reader needs to know which.

I was hoping that my comment about significant digits in Table 1 would lead the authors to consider significant figures throughout the paper. Please attend to that and reduce the unnecessary and over representation of precision/accuracy.

With access to the research data, I discovered that Figure 3 has major problems. The authors' equation to calculate R, the rate of relative change (i.e. a percent change) of N or P accumulation uses the term Si/S(i-1) as the numerator, which is then divided by the number of years to estimate an annual rate of relative change. Consider this example, if the N accumulates from 10 to 12 units over the period of 10 years, the equations reduces to (12/10)/10 or 0.12. The % change over the interval would be 20%, which would happen over 10 years, or a rate of % change of 2%/year. The problem with the equation becomes more extreme if the N accumulation is negative between successive stand ages (e.g. moss data). If the N goes from 12 to 10 units over 10 years, the equation reduces to (10/12)/10 or 0.0833. This does not reflect the negative change and seems a bit meaningless. The percent change would be -16.7% over 10 years or a negative 1.7%. The term Si/S(i-1) could be revised to [Si/S(i-1)]-1.

You will need to redo calculations in Table 3 as well.

In addition, the calculations for Figure 3 have been done for each quadrat, resulting in 15 data points (see S4 Supporting Information). The quadrats are not linked over the six time periods, so the accumulation data need to be averaged over the 3 quadrats for each time period before doing the rate of relative change analysis. This would result in 5 data points per graph. It would be helpful to label the different intervals associated with each point in the graph as early vs. Middle vs. late sequence rates of relative changes. This could provide additional interesting insights.

Sorry for the extensive comments on Figure 3, but the significance level included with the graphs is for the regression itself (slope significantly different from 0), not whether the slope is significantly different from 1. For this you need to establish the standard error of the coefficient (slope) and determine whether it is different from 1.0 at some desired p-value. For example, the given moss data (not correct as presented, but provided here as an example) lists the regression equation as:

y = 0.74x + 0.02 with p < 0.001

The standard error of the slope estimate (0.74) is 0.122. A p = 0.05 would be more appropriate here.

Reply: Following your suggestions, we have made a comprehensive revision of the equation, data, and Figure 3, as well as of the text in lines 242-250. Thank you very much for all your comments, which have greatly improved the manuscript.

Other specific comments

Fig. 2. Soil vs. Veg. proportions are reciprocals, so no need to show regressions for both. The regression is most likely driven by S1 for both N and P. The important ecological trend would be from S2 to S6, which looks weak, especially for N. I think this figure is more appropriate without the trend lines. The authors' interpretations in the text still stand, and showing a significance level does not really help. If statistical uncertainty is desirable to show, errors bars would be more appropriate than inferring a trend line that is not there.

Reply: Following your suggestions, we have modified Figure 2.

The Supporting Information S3 shows the same N value of 4.75 g/kg for S3 to S6 for A fabri. I'm not sure that Fig. 3a for "Tree" is legitimate. Separation of fine roots by species is usually difficult, and if an earlier publication has addressed how this difficulty was overcome, it should be cited (and quickly summarized).

Reply: We have modified the N value in Supporting Information S3, as there were some . errors in the data. Thank you very much for pointing out these mistakes. As for Figure 3a, the trees included were H. rhamnoides, Salix spp., P. purdomii, and A. fabri. Separation of fine roots by species have been reported in Wang et al. (2020). Roots were classified into fine (diameter <2 mm), medium (2-30 mm), and coarse (>30 mm). The procedure is explained in Supporting Information S2.

In Table 1:

The A – layer standard deviation seems low. Is this the Standard Deviation of the Mean or the simple standard deviation of the 4 samples?

Reply: The Standard Deviation of the A layer was calculated as the simple standard deviation of the three samples.

The A – layer bulk density is quite low (about 1/3 of a typical mineral soil). I believe that this is a granitic mineral soil (glacial till?) so if the A layer is this light, perhaps the boundary between the O and the A was indistinct. A previous study (Jiang et al. 2018) just collected 10 and 20 cm deep samples to estimate elemental concentrations.

From Jiang: "Due to the immaturity of soil development, the boundaries of the three horizons were inconspicuous, and none of the soils showed cambic(B), eluvial (E), or illuvial (Bs, Bh) horizons." Some explanation of how hard (or easy) this layer distinction was and how it was dealt with in the field would be helpful if the numbers in the table are to be believed. 

Reply: Regarding the classification of soil layers, there are some differences between our manuscript and that of Jiang, which are better explained in the present version. The soil profiles were O-horizon soil (organic soil), C-horizon soil (soil parent horizon), and bedrock, from top to bottom. The absence of a distinct A (mineral) horizon has been attributed to the short period of vegetation succession. However, the organic soil was divided into Oe (intermediate decomposed organic horizon) and Oa ( highly decomposed organic horizon). Litter was considered a plant sample without decomposition and thus not included in the soil stratification. Jiang divided the soil into three units: O, A, and C horizons, which represented litter layer, soil with dark brown color and humus enrichment, and soil parent material, respectively. The soil stratification section was revised; the “O layer” was changed to “Oe layer” and the “A layer” to “Oa layer”. Please check the lines 162-165.

Below I respond to the authors' responses in YELLOW.

Author's Response To Reviewer Comments

Replies to reviewer 1: Deane Wang:

Abstract/Introduction:

1. The abstract and introduction should emphasize the same important points. The abstract makes assertions about the role of the N:P ratio in limiting vegetation growth, while the third objective listed at the end of the introduction just list the desire to report on changes in the N:P ratio. It seems like the overriding goal is to better understand nutrient controls on ecosystem development through the description of N:P ratios, with the specific goal of assessing limiting factors.

Reply: One of the goals of this study was to better understand the role of N or P in the control of plant growth in the process of ecosystem development through the N: P ratio of vegetation. To clarify this point, We have revised the abstract and introduction, and the amendments are highlighted in the revised manuscript, such as in lines 41 and 104.

Good

Check the sentence with: "… which may the main sink for atmospheric N"

A "be" is missing.

Reply: Thank you for pointing out this mistake. We have revised the sentence. Please check line 41.

2.This work is part of a series of papers on the Hailuogou glacier retreat area, which should be acknowledged and cited in the Introduction. This helps the reader understand the context of the research, and may also help understand what work was done in connection with related research and what work was conducted specifically to answer the N:P ratio questions.

Reply: Previous researchers have done a lot of work in Hailuogou Glacier Retreat area, such as quantifying vegetation succession process, C accumulation process, soil P form, and effectiveness evaluation. In the introduction, We have summarized related research results, and the new text is highlighted in the revised manuscript. However, the dynamic changes of N and P in the whole ecosystem have not been studied in this primary succession sequence. Although Yang et al. [1] studied the dynamic changes of C and N in the ecosystem, they only studied the broad-leaved forest stage in this succession sequence, and did not study the coniferous forest stage in the late succession stage, so it is incomplete for the forest succession process. Therefore, based on the previous research results, we studied the dynamic process of the complete vegetation succession sequence for N and P. We revised this for clarity in lines 88-99.

See comments on the Introduction above. While these new lines are helpful, this Introduction still avoids reviewing the literature on N:P at your site and the implications for understanding nutrient limitations on ecosystem development. Your paper provides a whole ecosystem data set on N:P which complements earlier research attempting to use N:P to understand ecosystem development. Be explicit about the new perspective you are offering the research community. Later, in your discussion, you can be honest about whether you think your new ecosystem-level data has helped establish N as the ongoing limitation (or not).

Reply: We have modified both the Introduction and Discussion sections, and further explained the significance of our research. Thank you very much for your comments.

METHODS

1.A key paper to describe vegetation methods might be: Yang D, Luo J, She J, Tang R (2015) Dynamics of vegetation biomass along the chronosequence in Hailuogou glacier retreated area, Mt. Gongga. Ecology and Environmental Sciences 24:1843–1850 (in Chinese with English abstract). However, I could not access this paper. If the chronosequence and the biomass sampling are the same, then perhaps this material could be included in the supplementary material. (I could also not access the supplementary material for this paper - Yang et al.).

Reply: The chronosequence and the biomass sampling in the present study were the same as those reported in Yang’ paper[2]. We set up six sampling sites in the Hailuogou glacier retreat area (glacier retreat time: 2000, 1980, 1970, 1958, 1930, and 1890). Specific biomass information has been added to the supplementary material.

This is helpful. See comment above about clarifying the biomass data source, which can simplify your need to detail biomass methods and materials in this paper. If you feel that the other biomass paper is less accessible to English language readers, you can provide the details in another Supporting Information section.

Reply: Detailed information on biomass is presented in Supporting Information S2.

2.Currently, the Methods section omits much detail, and if this is because the work was related to other work, this should be described (and cited). If, for example, the sampling was done specifically for this paper, then a great deal more information needs to be provided. For example, were the tree subsamples and the allometric equations developed from the same trees? If the vegetation was sampled for this research, how were the replicate samples distributed within a site? Were they randomly selected? Were shrub and ground cover samples nested within the tree samples? Were the soil samples nested in any of the other sample locations? The high spatial variability of soils and vegetation is generally a challenge for ecosystem research and some acknowledgement of this and how this team approached this problem is useful for future investigators.

Reply: The sampling of the vegetation biomass has been described in detail elsewhere [2]. Briefly, six sampling sites were chosen for vegetation sampling basing on the glacier retreat time (2000, 1980, 1970, 1958, 1930, and 1890). At each site, three quadrats of 10 m × 10 m were established. All trees with a diameter at breast height (DBH; 1.3 m height aboveground) of >2 cm were inventoried in each quadrat. The species name, DBH, total height, and geographical coordinates were recorded in each quadrat. Based on the information reported by Yang et al.[2], Liu et al.[3] established the dominant tree species (H. rhamnoides, Salix spp., P. purdomii, and A. fabri). Allometric equations are commonly used to estimate tree and stand biomass from easily measured dendrometric variables such as DBH or height. Briefly, an additive allometric equation for tree biomass components against DBH and H was as follows:

In(y) = a + b*In(DBH^3/H)

Detailed discussion of the four tree species in this study can be found in Liu et al. [3].

The shrub, herb, and moss quadrats were nested within the tree quadrat. In each quadrat of 10 m × 10 m, we established a quadrat of 5 m × 5 m to harvest the shrub biomass, and 1 m × 1 m subplots on the forest floor at each site were prepared, and the mosses and grasses were collected.

The soil samples were nested in each quadrat of 10 m × 10 m. We determined the sampling plot of the soil profile at each sampling site according to terrain, slope, and other conditions, and then a 0.5 × 0.5 m subplot at each site was established to sample litter and soil. We revised this for clarity in lines 130-143.

See comment directly above. Again, now that I am aware of the relationships of these papers, it might be more efficient to just state the biomass data were obtained from the earlier paper and research. Presumably this is a peer reviewed paper and the specific methods have passed that review.

Reply: Following your comments, we have revised the corresponding section.

3.The context of this research is also a bit puzzling relative to:

Yang et al. (2014) Dynamics of carbon and nitrogen accumulation and C:N stoichiometry in a deciduous broadleaf forest of deglaciated terrain in the eastern Tibetan Plateau. Forest Ecology and Management 312: 10–1. That work indicates its chronosequence location as "This study was conducted in the foreland of the Hailuogou glacier (101.99°E, 29.57°N, 2990 m a.s.l.)" the whole successional chronosequence, the Populus purdomii forest acts as the broadleaf deciduous stage, which is a band in the valley bottom at elevation from 2844 m to 2950 m." This paper (under review) also list the elevation: "... forest primary succession sequence of approximately 2 km was formed in the Hailuogou glacier retreat area at an altitude of 2800– 2970 m “with similar Lat/Long coordinates. If these chronosequences are the same, the Methods and perhaps the Introduction, should indicate this. Because the C:N stoichiometry over a chronosequence would require essentially the same methods as a N:P stoichiometry, I would think that these were related studies. This should be clarified.

Reply: Yang et al.[1]and the present study investigated the same forest primary succession sequence in the Hailuogou glacier retreat area, but there were many differences. First, The glacier retreat year was determined by time summary of the ecesis interval of pioneering tree species (i.e., time between the glacier retreat and tree seedlings germinating) and the maximum tree age in the glacier retreated areas [2, 4–6], and through tree rings for correction. Yang et al. [1] was based on the terrain age every 10 years, which may not represent the exact year of glacier retreat.

Second, which is the most important difference, Yang et al. [1] only studied the broad-leaved forest stages of this forest primary succession sequence. However, this forest primary succession sequence in the Hailuogou glacier retreat area is the complete successional chronosequences from the pioneer vegetation (Astragalus souliei Simps, H.rhamnoides) to broad-leaved forest stages (P. purdomii, H. rhamnoides, Salix spp.), and then to the climax community (Abies fabri (Mast.) Craib and Picea asperata Mast.). We studied the whole successional chronosequences from the pioneer vegetation to the climax community (including the broad-leaved forest stages). Therefore, Yang et al. [1]and the present study investigated the same area of forest primary succession but not the same successional chronosequences. Yang et al. [1] only studied the broad-leaved forest stage of this forest primary succession (the glacier retreated for about 60 years), whereas we studied the complete forest primary succession sequence (the glacier retreated for about 125 years).

Finally, 2990 m is the glacier forehead, 2844–2950 m [1] is the succession stage for P. purdomii forest, which acts as the broadleaf deciduous stage, 2800–2970 m (the present study) is the whole succession stage from pioneer plants to climax communities. Because of the different chronosequences, the lat/long coordinates and altitude of Yang et al. [1]and the present study are also different.

Revisions to your Introduction could alleviate possible confusion that the reader may have (myself as an example). You do have the obligation in your discussion to highlight any differences about succession that the two articles may have, for example, differences in biomass accumulation rates (which would translate to nutrient accumulations rates).

Reply: We have modified the Introduction (please check lines 95-100) following your comments. Thank you very much. We believe the reader will now have a clearer understanding of the research background.

4.Were the biomass estimates separate estimates from that of Yang et al. (2014)? If so, some comparison with those biomass and N accumulation rates should be presented in the Results and Discussion.

Reply: The biomass estimates were different from the reports of Yang et al. [1], but the same as those from Yang et al. [2] Specific biomass estimation methods and related research results have been explained above. We have also included comparisons of biomass and N accumulation rates in the Results and Discussion sections, and new contents are highlighted in red in the revised manuscript, such as in lines 245-250.

New content comparing biomass estimates from Yang et al. (2014) and Yang et al. (2015) was not apparent around lines 245-250. For the overlap in chronosequence years, I would assume similar accumulation rates, but it would be interesting to see the data.

Reply: Unfortunately, no additional information on biomass accumulation rate could be retrieved from Yang et al. (2014), meaning it is not possible to compare data. As so, we have removed this section from the present version. However, in future research, we will seriously consider this question. Thank you very much for your comments.

5.The regression statistics used (line 161) do not seem like they are appropriate for the

statements made about the slope being different from one. Without the data, I can only do a visual estimate, but the significance level of p<0.001 seems inappropriate for the presented data, especially for tree, herb and the A layer. A more detailed statistical explanation is needed.

Reply: All statistical analyses were conducted using SPSS 21.0. The relationships between the relative change rates of N and P were analyzed by linear regression. A more detailed statistical explanation was added to the supplementary material.

I could not find additional statistical explanation in the Supporting Information, but I see that the linear regressions were run in Excel. The reported p values are for the linear regression (slope different from 0), and I have commented on these statistics in my Overall Comments.

Reply: Following your comments, we have revised this section.

RESULTS and DISCUSSION

1.Many factors limit vegetation growth, both individually and in concert with each other. Moisture, sunlight, temperature, nutrients, etc. can result in the observed rate of biomass accumulation. The reported rates of biomass accumulation across this chronosequence seem comparable with other estimates of forest growth rates in similar climates. These glacial ecosystems do not appear to be growth-limited in any unusual ways. Accepting that this study seeks only to examine N and P limitations, it is a fair question to wonder about N vs. P limitations. However, I think it is useful for the authors to present this context of many limitations, and their particular interest in N and P, to their readers.

Reply: In the Hailuogou glacier retreat area, which is about 2000 m in length and 200 m in width, it is generally believed that the moisture, sunlight, and temperature control the plant belt spectrum and forest line formation changes little, which cannot be the main reason for the control of the emergence and succession of vegetation in bare land. Therefore, the nutrient elements necessary for plant growth may be the main reason for the development of vegetation. In this manuscript, the limitation of N and P in this succession sequence was examined through the distribution and accumulation characteristics of N and P, and the N:P ratio. The overriding goal was to better understand nutrient controls on ecosystem development.

What you say above makes sense. I do think you do need to more explicitly address why you think your ecosystem-level data on N:P ratios improves our previous understanding of nutrient limitations based on what others have reported for this area.

Reply: Thank you for your suggestion. A section detailing our conclusions on ecosystem-level N:P ration has been added as shown below.

Three factors regarding the ecosystem-level of N:P ratios may help establishing N as limiting nutrient. First, the mass ratio of the leaf only reflects the allocation of N and P at the organ-level, while the pool ratio of N:P reflects the allocation of both nutrients at the ecosystem-level. Second, the ecosystem-level of N:P ratios can reflect the relationship between N and P in the different ecosystem components and clarify their interactions along the succession. In the present study, we calculated the rates of relative N and P changes of each ecosystem component based on N and P pools and found that the N:P ratio in vegetation decreased slightly and was lower than that of organic soil. This was due to the higher P accumulation rate compared with the N accumulation rate. The influence of accumulation rates between N and P on N:P ratio is not reflected on the mass ratio. Finally, the total vegetation N:P ratio ranged from 8.4 to 13.3 in the present study. After 15 y of recession, the N:P ratio of total vegetation might have peaked in this succession sequence due to the N-fixing plants at this site. As leguminous plants withdrew from the community, the N:P ratio in vegetation dropped to a constant value along this sequence, reflecting the long-term and stable corresponding values of N and P in vegetation to a state of equilibrium. This is the embodiment of ecosystem stability, thus promoting the positive succession of vegetation. Therefore, in the long-term development of the ecosystem, N was considered as the main limiting nutrient.

2.As an alternative interpretation to their conclusion of N limitation, I might suggest that the accumulation of N in the soil demonstrates the build up of "extra" N. "Tight" cycling of N has been demonstrated in many forested ecosystems, such that the pool of organic N in the forest floor and the soils remains relatively constant over successional time. The absence of this "tight" cycling of N in this primary succession could be interpreted as an indication of the relative availability of N. P on the other hand, is much more stable over the chronosequence in the litter, O and A layers, possibly suggesting that the continuing requirement of P by accumulating biomass takes up any new P from decomposing organic matter while accessing the continuing release of P from primary (and possibly secondary) minerals to fine roots and mycorrhizae.

Reply: The reviewer's interpretation provides another way to explain the dynamic changes in N and P from the source and cycle of N and P. The N concentration in the original bare land of Hailuogou was almost 0 [2]. Through biological N fixation, atmospheric N deposition, and litter decomposition, the N in the surface soil is continuously accumulated, and the nutrient conditions are improved. Because the glacial retreat of Hailuogou is only around 120 years old, it is still at a "young stage" relative to the soil age, and no complete soil profile has been formed. Although the succession of forests has formed the climax community, soil development continues. Therefore, N in the soil may continue to accumulate. In this succession sequence, forest succession and the dynamics of soil P are an interactive process, especially after the formation of a coniferous forest climax community dominated by A. fabri and P. brachytyla. On the one hand, these coniferous forest trees have a stronger ability to secrete organic acids; on the other hand, the acidity of needle leaf litter is lower, which makes the soil pH significantly lower in the sampling site of glacier retreat for 125 years (pH = 4.4). However, the rapid decrease in soil pH results in the accelerated release rate of primary mineral P, and the P mineralization pool ( the supply of available P from mineralization of organic P in the organic layer) was ~2.9 times that of the P requirement [7]. This is consistent with the reviewer's explanation.

3.In the Discussion the authors state: "Generally, a N:P ratio <14 in leaves indicates that plant growth is limited by N, whereas a N:P ratio >16 indicates that plant growth is limited by P." (line 307) So it is curious that they do not report their own N:P ratio in leaves as a comparison. Total vegetation N:P ratios are a weaker absolute indicator of nutrient limitations due to the various N:P ratios in different species and different vegetation tissues. They cite (line 311) a reported range from "<10 for N limitation" to ">20 for P" across a broad range of vegetation. Their reported ratio is just at the boundary of 10, which does not make a strong case for N limitation.

Reply: In the corresponding part of the manuscript, I added content related to the N:P ratio of leaves on each dominant tree species and the whole tree layer. The results are as follows:

Table 1. N: P ratio of leaves in different species and tree layers

N:P ratio in leaves

Sampling site Glacier retreat year H. rhamnoides Salix spp. P. purdomii A. fabri Tree

S1 15y 19.63 4.16 13.68 10.99

S2 35y 18.16 5.00 15.42 10.14

S3 45y 17.79 10.14 14.81 10.45 14.30

S4 57y 14.89 8.06 12.84 10.01 12.68

S5 85y 12.61 10.37 10.42

S6 125y 10.48 10.48

It can be seen from the Table 1 that the N:P ratios of different dominant trees were quite different. As a N-fixing plant, the N concentration in H. rhamnoides leaves was high, so the N:P ratio in its leaf was higher than that in other dominant tree species, whereas the mean P concentration in Salix spp. leaves was 4.17 g kg-1, which was higher than that of the other tree species. Therefore, the N:P ratio in its leaves was extremely low. Due to the different absorption of N and P by different tree species, the concentration of N and P in leaves of different tree species were different. Therefore, when we consider the N: P ratio of vegetation leaves in each sample plot, we consider the N: P ratio of tree layer leaves after weighting. From S1 to S6 sites, the N:P ratio in tree leaves ranged from 10.14 to 14.30, except in the S3 site, and the N: P ratio of other sites was significantly lower than 14 (N:P ratio <14 in leaves indicates that plant

growth is limited by N). In addition, the N: P ratio of leaves of A. fabri, for which the biomass of the ecosystem at the S6 site was >89% [2], was also found to be lower than 14. Therefore, only from the N:P ratio of leaves did we tend to consider that the growth of plants was probably limited by N.

We mainly studied the dynamics changes in N and P pools in the whole primary succession and discussed the relationship between N and P in the ecosystem. Therefore, when we study N:P stoichiometry of ecosystems, we should consider carefully the pools of N and P [8]. Thus, a given N:P ratio in the present study may refer to total N and P pools, and this can reflect the N: P ratio of the whole vegetation layer. Thus, we describe a reported range from <10 for N limitation to >20 for P limitation across a broad range of vegetation. The N:P ratio for vegetation ranged from 8.35 to 13.25 (S1: 13.25, S2: 9.66, S3: 9.79, S4: 9.01, S5: 8.35, S6: 10.00) and remained constant after the S2 site.

The N:P ratio in different study areas, ecosystems, and vegetation types might vary greatly. Therefore, most scholars accept the view of ecosystems being N-limited at a low N:P ratio and P-limited at a high N:P ratio. Although the N:P ratio of leaves and vegetation layer are different to some extent, these ratio were lower than the values currently thought to reflect N limitation. The present study sought only to examine N and P limitations through their N:P ratio. It is indeed difficult to assess N or P limitation to plants by only relying on stoichiometry ratios of N and P. We need more evidence to support this conclusion. Therefore, combined with previous studies, I presented more evidence, including soil nutrient supply and nutrient-addition experiments, to support that the growth of plants in the Hailuogou glacier retreat area may limited by N. First, Zhou et al. [9-11]carried out much research on soil P in the Hailuogou glacier chronosequence. Their results showed that: at 120-y-old-site, the Pavailable pool in the organic layer and 0–6 cm mineral soils was 27.0 kg ha−1 and ~5.3 times the annual plant P requirement; the Pmineralization pool, representing the supply of available P from mineralization of organic P in the organic layer, was ~2.9 times the P requirement; the P weathering pool was 7.5 kg hm−2 y−1 and was higher than the P requirement. These results suggest that the current P pools can offer enough P for the growth of the ecosystem. Second, although it is difficult to carry out a nutrient-addition experiments in this primary succession sequence, a N-addition experiment on A. fabri seedings may be helpful to understand nutrient controls on plant growth. Yang et al.[12]conducted such an experiment in the Gongga Mountain observation station, which is ~1 km away from the Hailuogou glacier retreat area. They found that the total biomass, leaf dry weigh, leaf mass ratio, leaf N and P concentration, and leaf N:P ratio of the A. fabri seedlings increased by 11.29%, 46.70%, 41.40%, 37.30%, 22.33%, and 6.43%, respectively, after 2 years addition of N (50 kg N hm−2 y−1), indicating that the growth of A. fabri seedlings was probably limited by N. Furthermore, our study also showed that the biomass accumulation rate may be more positively correlated with the accumulation rate of N.

Therefore, from the results of N:P stoichiometric of vegetation, soil P supply, and N-addition experiments, we suggest that the plant growth in the Hailuogou glacier succession sequence may be limited by N. The text has been revised to address this in lines 257-265, and 387-407.

This is helpful. You might want to emphasize that you believe that "N limitation of ecosystem growth continues to be a factor after 120 year of ecosystem development." Would you speculate from your knowledge of the literature of A. fabri cecosystems when the N limitation would start to become less of a factor? This would be a helpful conjecture for directing future research on these questions.

Reply: After 125 y of recession, the absence of a distinct A (mineral) horizon is attributed to the immaturity of soil development. We hypothesized that with soil development, especially after the complete formation of the soil profile, the N-fixation of microorganisms in the soil was further enhanced, thereby alleviating N limitation.

4.Line 41 of the Abstract states: "... N was the main limiting factor for plant growth in this sequence." Given that an abstract may be the only part of the research that some scientists may read, I would suggest that this conclusion needs to be stated more carefully, perhaps with reservations.

Reply: We have modified the corresponding statements in the article and the amendments are highlighted in red in the revised manuscript, such as lines in 41, 104, 376, 384, 406, 421.

Good

SPECIFIC COMMENTS

line 98: If an actual average can not be calculated from recorded data, then two to four

significant figures are not needed to give the reader an approximate estimate for temperature and rainfall. In this case ~4 Deg C and ~2000 mm rain would suffice.

Reply: We have revised the corresponding part of the paper to make the expression more scientific and the amendments are highlighted in red in the revised manuscript, such as lines in 110.

Good

line 120: all of the numbers in Table 1 should include at most two significant figures. The estimates are not more precise than that, and having fewer digits to examine makes it easier for the reader.

Reply: We have revised the numbers in Table 1 to clarify the results for the reader.

Good. Refer to comment about significant figures

line 121: Tree biomass was estimated using the allometric equations reported by Liu (2019). Are these the same equations reported by: X Zhong, N Wu, J Luo, K Yin, Y Tang, Z Pan -Chengdu Science and Technology …, 1997. Researches of the forest ecosystems on Gongga Mountains.

Reply: They are not the same equations. Liu et al.[3] estimates were based on the measured biomass of the main tree species in the primary forest succession in the Hailuogou glacier retreat area, the total biomass of the trees and the biomass of different components (such as branches, leaves, trunks and roots) together with the breast diameter and tree height. The allometric equations were for four common tree species: A. fabri, P. purdomii, H. rhamnoides, and Salix spp.

If you feel these equations are better than previously used equations, you might say that. The allometric equations might be included in the Supporting Information as they are not easily accessible outside of China.

line 125: The difference between "herbs" and "ground cover" is not clear.

Reply: "Herbs" are herbaceous plants, while "ground cover" refers to moss. We revised the "ground cover" to "moss" in the manuscript to clarify this point. We revised this for clarity in lines 141, 211, 255, 238, 246. 

line 143: "The thickness and bulk density of each soil layer were then measured using a measuring tape and a cylindrical tube, respectively." Soil is notoriously hard to sample because of the inherent spatial variation. It would be helpful to include the N (sample size at each site). Table 1 also needs to indicate the unit of the variation (one standard deviation, one standard error (standard deviation of the mean), two standard errors?). Pit sampling (TG Huntington, DF Ryan, et al. 1988 Estimating soil nitrogen and carbon pools in a northern hardwood forest ecosystem. in Soil Science Society of Am.) provides a more accurate bulk density and nutrient concentration estimate. For ease of examining the data, it would be very helpful to use two/three significant figures, especially given the high variation that seems apparent, e.g. 120.80 +- 13.89 should probably be reported as 120 +- 14; 6.43 +-0.53 is probably best as 6.4 +- 0.5.

Reply: We have revised the data in Table 1 to include data about biomass, surface soil pH, thickness, and bulk density, including the mean value and one standard deviation.

line 174: "vegetation N pool sharply increased" The word "sharply" does not seem appropriate here as the accumulation rate of N over the 125 years appears linear. "Sharply" is used again on line 229.

Reply: We have removed the word "sharply" regarding the vegetation N pool, such as lines in 207 and 291. 

line 201: The rates of relative N and P changes in Figure 3 are not consistent with the explained methods. Equation (1) line 157 shows the calculation of rate of relative change for each site (i = 1– 6). Figure 3 shows more than 6 points. The construction of Figure 3 needs to be explained.

Reply: Figure 3 shows the line regression about the relative change rates of N and P. We divided the ecosystem into six components: tree, shrub, herb, moss, O layer, and A layer. For each component, the rate of relative N or P change was calculated as follows: the relative N or P changes were calculated as the N or P pool at the current age stage divided by that at a previous age stage. Then the rate of relative N or P changes was obtained by the relative N or P pool change divided by the age interval between two adjacent age stages [1, 13].For example, rate of relative N or P change in trees: where Rtree is the rate of relative N or P change in trees; S represents the tree N or P pool; T represents the time of glacier retreat; and i is each site (i = 1–6). Each site had three replicates. More detailed information was added to the supplementary material.

See my comment above about Figure 3.

line 202: The statistics associated with Figure 3 indicate a significant correlation between N and P (i.e. a slope significantly different than 0). However, in order to make a statement about the importance of the slope being different than 1.0, the 95% confidence bounds on the slope estimate (e.g. slope of 1.26 for trees) needs to be provided. Also Figure 3 shows a slope of 1.26 while line 206 states a slope of 1.32. Other inconsistencies between the text and Figure 3 are also presented.

Reply: We have revised the mistakes pointed out by the reviewer. The 95% confidence bounds on the slope were: tree: 1.265 ± 0.245, shrub: 0.848 ± 0.066, herb: 0.874 ± 0.323, moss: 0.744 ± 0.264, O layer: 0.735 ± 0.343, and A layer: 0.873 ± 0.207. We revised the text to address this point in line 241.

You rely on these slope differences for your statements about N vs. P limitations. You should include the statistics above in the text or in the figures. Your reader needs to know the basis of you statements of statistical significance. You probably should not include the linear regression data on Figure 3 as the positive correlation between N and P accumulation is self-evident in an aggrading ecosystem.

Reply: We have added the corresponding data of statistical analysis in Figure 3.

line 299: "also showed that the rate of relative N accumulation was faster than that of P in surface soil." Without knowing if the reported slopes were significant or not (see comment above, line 202), the results of this study may only "suggest." However, the point here should probably be about which processes are causal and which observations are just incidental to those processes. A higher relative N accumulation rate WILL result in a change in N:P ratio, in all cases. The more salient question is how N and P cycling are changing (relative to each other) as the forest stand matures.

Reply: The slope of regression for the rate of relative N and P changes was compared with 1, and we were able to detect how the accumulation of N changed with P accumulation along the successional gradients. Thus, we can understand the relationship between N and P for different ecosystem components to clearly understand N–P interactions along the successional gradients. The threshold p value of regression in the O layer and A layer was = 0.001 and <0.001, respectively, indicating that the results were significant. However, making a causal claim without knowing which processes are causal and which observations are incidental to those

processes is not appropriate. Therefore, We have revised the manuscript accordingly with more appropriate interpretation or hypotheses. In the corresponding part of the paper, We revised “showed” to “suggested”.

The more salient question raised by expert is the focus of our next research. Examining the nutrient cycling is to understand the N and P utilization, circulation, and transmission among different components of the ecosystem. Especially in different succession stages, the different dominant species have different ways of nutrient cycling, which will lead to differences in N, P concentration and N:P ratio.

This is helpful, but the p value in the reply for the slope difference from 1 is NOT correct (see comments under Overall Comments).

line 301: In this primary successional sequence it seems unlikely that P has weathered out of the rocks and is less available at the end of the chronosequence.

Reply: Zhou et al. [11]reported that the weathering processes in the Hailuogou glacier

chronosequence are rapid due to fast vegetation succession, higher temperatures, and relatively high precipitation. In addition, Zhou et al. [9] reported the average rate of weathering of primary mineral phosphate (RLP) in this chronosequence. The average RLP (14.1 mmol m−2 y−1) in the Hailuogou glacier chronosequence was ~47 times higher than the global rate of P release. Especially at the 120-y-old site, the RLP was significantly higher than at sites with similar ages in temperate and subtropical zones. Zhou et al. [11] reported the changes in soil P speciation along this chronosequence and indicated that the concentration of bioavailable P in surface soil showed a trend of increasing, with 5–11.5% of total soil P, and the stocks of bioavailable P were greater than the annual P requirement of vegetation. In this primary successional sequence, P

had not only weathered out of the rocks but was also available at the end of the chronosequence.

I think that we are agreeing here.

line 316: "owing to the high N level of N-fixing" adding "perhaps due to" would be a fairer statement of likelihood. This study did not establish the biogeochemical role of N-fixing microbes and plants at these sites. Making a causal claim about what particular biogeochemical process alters the relative uptake of nutrients is therefore not appropriate as a statement, and more appropriate as an interpretation or hypothesis.

Reply: Leguminous plants (such as Astragalus adsurgens Pall. and Astragalus souliei Simps) and H. rhamnoides are the dominant species in the S1 site. As N-fixing plants, their N concentration was higher than that of other species, which may lead to the higher N:P ratios. However, we did not establish the biogeochemical role of N-fixing microbes and plants at this site, so making a causal claim is not appropriate. We revised “owing to” to “ perhaps due to”, such as line in 379.

line 326: "This increase in N and decrease in P may shift the factor limiting plant growth from N to P after several hundred years. Therefore, the dynamics of the N:P ratio in this forest primary succession needs further study." Successional changes in relative nutrient availability are important to document, and this 125 year sequence provides a good example of nutrient dynamics in a chronosequence. However, the N limitation assertion is an hypothesis, and the authors should be careful in how they state their interpretations. Given the various ratios of N:P they report at their sites, the N limitation is possible, but not conclusive.

Reply: We have revised the manuscript according to this comment to offer interpretation or hypothesis statements, which are more appropriate, such as lines in 41, 104, 376, 384, 406, 421.

line 339: "whereas the N:P ratio in vegetation maintained a constant low level due to the tree layer having a rapid P accumulation rate compared with N;" "higher" rather than "rapid" is probably more appropriate here. Plants and ecosystem accumulate nutrients according to their needs, and we can only infer limitation if we do not determine this experimentally (e.g. with a nutrient addition experiment).

Reply: In the corresponding part of the manuscript, We revised “rapid ” to “ higher”, such as lines in 240, 241, 350, 358, 382.

---

## [Editor Report · Decision Letter 2]

20 Jan 2021

Dynamics of nitrogen  and phosphorus  accumulation and their  stoichiometry along a chronosequence of forest primary succession in the Hailuogou Glacier retreat area, eastern Tibetan Plateau

PONE-D-20-29430R2

Dear Dr. Peng,

We’re pleased to inform you that your manuscript has been judged scientifically suitable for publication and will be formally accepted for publication once it meets all outstanding technical requirements.

Kind regards,

Dafeng Hui, Ph.D.

Academic Editor

PLOS ONE

Additional Editor Comments (optional):

The authors have adequately addressed most concerns raised by the reviewer. My decision is Accept.
---

## [Editor Report · Acceptance letter]

22 Jan 2021

PONE-D-20-29430R2 

Dynamics of nitrogen and phosphorus accumulation and their stoichiometry along a chronosequence of forest primary succession in the Hailuogou Glacier retreat area, eastern Tibetan Plateau 

Dear Dr. Peng:

I'm pleased to inform you that your manuscript has been deemed suitable for publication in PLOS ONE. Congratulations! Your manuscript is now with our production department. 

Kind regards, 

on behalf of

Dr. Dafeng Hui 

Academic Editor

PLOS ONE